



# Assessing Variability of Wind Speed: Comparison and Validation of 27 Methodologies

Joseph C. Y. Lee[1,2], M. Jason Fields[1] and Julie K. Lundquist[1,2]

[1]National Renewable Energy Laboratory, Golden, Colorado 80401, USA

[2]Department of Atmospheric and Oceanic Sciences, University of Colorado Boulder, Boulder, Colorado 80309, USA

*Correspondence to*: Joseph C. Y. Lee (Joseph.Lee@nrel.gov)

**Abstract.** Because wind resources vary from year to year, the inter-monthly and inter-annual variability (IAV) of wind speed is a key component of the overall uncertainty in the wind resource assessment

process thereby causing challenges to wind-farm operators and owners. We present a critical assessment of several common approaches for calculating variability by applying each of the methods to the same 37-year monthly wind-speed and energy-production time series to highlight the differences between these methods. We then assess the accuracy of the variability calculations by correlating the wind-speed variability estimates to the variabilities of actual wind-farm energy production. We recommend the

Robust Coefficient of Variation (RCoV) for systematically estimating variability, and we underscore its advantages as well as the importance of using a statistically robust and resistant method. Using normalized spread metrics, including RCoV, high variability of monthly mean wind speeds at a location effectively denotes strong fluctuations of monthly total energy generations, and vice versa. Meanwhile, the wind-speed IAVs computed with annual-mean data fail to adequately represent energy-production IAVs of

wind farms. Finally, we find that estimates of energy-generation variability require $10 \pm 3$ years of monthly mean wind-speed records to achieve 90% statistical confidence. This paper also provides guidance on the spatial distribution of wind-speed RCoV.

**Keywords:** Inter-annual variability, Statistics, Uncertainty quantification, Variability, Wind resource

assessment



## 1 Introduction

The P50, a widely used parameter in the wind energy industry, is an estimate of the threshold of annual energy production of a wind farm that is expected to exceed 50% of the time (Clifton et al., 2016). The P50 is usually estimated to apply over the lifetime of a wind farm, typically 20 years. To estimate

P50 in the wind resource assessment process, a single percentage value is usually assigned to represent the uncertainty for the desired certain time period at a wind site (Brower, 2012). The inter-annual variability (IAV) of wind resources, along with site measurements and wind plant performance, is an important component in the overall uncertainty in power production (Clifton et al., 2016; Klink, 2002; Lackner et al., 2008; Pryor et al., 2006). The IAV is also incorporated in the measure-correlate-predict

(MCP) process (Lackner et al., 2008), which usually considers wind measurements spanning less than 2 years.

Analysts and researchers use numerous metrics to quantify wind-speed variability, and the most common method is standard deviation (σ). For instance, the variability in historical or future wind resources is often represented as the σ from the annual-mean wind speed of a certain location (Brower,

2012). As wind-turbine power generation is a function of wind speed, the variability of wind resources has important implications on resultant long-term energy production. Financially, when the wind resource is projected to fluctuate more from year to year (Hdidouan and Staffell, 2017), the levelized cost of wind energy increases as well.

Because the profitability of wind farms depends on wind variability, past research has explored the

implications of inter-annual and long-term variability in wind energy. Pryor et al. (2009) analyse trends of annual wind speed and IAV, without explicitly quantifying IAV values. Archer and Jacobson (2013) evaluate the seasonal variability of wind-energy capacity factor. Lee et al. (2018) assess the spatial discrepancies between wind-speed variabilities of different temporal scales, from hourly mean to annual-mean data. Bett et al. (2013) use standard deviation (σ) and Weibull parameters to assess the wind

variability in Europe. Extreme event analysis also offers another perspective to assess variability. For example, Cannon et al. (2015) examine extreme wind-energy generation events via reanalysis data and discuss the associated seasonal and inter-annual variability qualitatively. Leahy and McKeogh (2013) also quantify the return periods of multi-week wind droughts.



To quantify variability, the normalized standard deviation or the Coefficient of Variation (CoV), the

σ divided by the mean of a time series, is a commonly used tool. Justus et al. (1979) calculated and

compared the CoVs of monthly and annual wind speeds at different sites across the United States. Baker

et al. (1990) quantified inter-annual and inter-seasonal variations of both wind speed and energy

production at three locations in the Pacific Northwest. They found the annual CoVs ranged from 4% to

10%, matching the conclusions from Justus et al. (1979). Recently, Li et al. (2010) calculate hub-height

wind-speed variance and σ of 30 years to spatially evaluate seasonal and inter-annual variability in the

Great Lakes region. Bodini et al. (2016) estimate the IAV of wind resources with a modified version of

CoV, using observed meteorological data in Canada. As the sample period increases, the IAVs of most

sites gradually increase, averaging 5 to 6% among the chosen sites (Bodini et al., 2016). Krakauer and

Cohan (2017) correlate the CoVs of monthly mean wind speeds with different climate oscillation indices,

and find the global mean CoV at 8%. In addition to characterizing wind speed, the metric is also used to

evaluate the benefits of grid integration. For example, Rose and Apt (2015) conclude the inter-annual

CoV of aggregate wind-energy generation in the central U.S. at 3 ±0.1%, much smaller than that of

individual wind plants between 5.4% and 12%, ±4.2%.

Aside from CoV, other metrics representing the spread of data have also been chosen to estimate

variability in the literature. For example, the Robust Coefficient of Variation (RCoV) normalizes the

median absolute deviation (MAD) with the median. Gunturu and Schlosser (2012) quantify the spatial

RCoV of wind-power density in the U.S. and demonstrate that the regions east of the Rockies, especially

the Plains, generally have weaker variability and higher availability of wind resources. Seasonality index,

originally used in Walsh and Lawler (1981) for precipitation purpose, is another measure to express

variability. Seasonality index is defined as the sum of the absolute deviations of monthly averages from

the annual mean, normalized with the annual mean. Chen et al. (2013) use the seasonality index to assess

the inter-annual trend and the variability of wind speed in China, and they relate wind-speed IAVs to

climate oscillations.

Alternative variability metrics emphasize the long-term trends via contrasting wind speeds of different

periods. The "wind index", used in Pryor et al. (2006) and Pryor and Barthelmie (2010), is a ratio of wind



speeds of a reference period and an analysis period. An entirely different wind index evaluated in Watson et al. (2015) is a ratio of spatially-averaged wind speeds during two different periods.

Despite the importance of long-term variability, the wind-energy industry lacks a systematic method to quantify this uncertainty. As various metrics to assess variability exist, a comprehensive comparison
of measures is necessary. Therefore, the goal of this study is to evaluate various methods of estimating inter-monthly and inter-annual variability in a reliable way using a comprehensive long-term database. Specifically, our objective is to determine an optimal metric or metrics for relating wind-speed variability to energy-production variability. We describe the wind-speed and energy-generation data, the methodology and the chosen variability metrics in Section 2. We evaluate different variability measures
via two case studies in Section 3. We also contrast the results computed from monthly mean and annual-mean data, and we illustrate the spatial distribution of wind-speed variability in Section 3. We then recommend the best practice in using the ideal method in Section 4. We focus on the applicability of imposing such metrics to quantify the variabilities of wind-speeds and wind-energy productions.

## 2 Data and Methodology

### 2.1 Wind and Energy Data

In this study, we use a 37-year time series of monthly mean wind speed and monthly total wind-energy production in the Contiguous United States (CONUS). For wind speed, we use hourly horizontal wind components in NASA's Modern-Era Retrospective Analysis for Research and Applications, Version 2 (MERRA-2) reanalysis dataset (Gelaro et al., 2017; Global Modeling and Assimilation Office (GMAO),
2015) from 1980 to 2016. We use these components to derive the monthly mean wind speed at 80 m above the surface, to represent hub height in this study, via the power law (1) and the hypsometric equation (2):

$$\frac{u(z_2)}{u(z_1)} = \left(\frac{z_2}{z_1}\right)^{\alpha},$$ (1)

$$z_2 - z_1 = R_d \overline{T} ln\left(\frac{p_2}{p_1}\right).$$ (2)

In (1), $u(z_1)$ and $u(z_2)$ are the horizontal wind speeds, at heights $z_1$ and $z_2$, in which wind speeds are the square root of the sum of squared horizontal wind components, and $\alpha$ is the shear exponent; in (2), $R_d$ is



the dry air gas constant, $\bar{T}$ is the average temperature between levels $z_1$ and $z_2$, and $p_1$ and $p_2$ are the atmospheric pressures at $z_1$ and $z_2$. In most grid cells, we use the MERRA-2 meteorological output at 10 and 50 m above the surface to calculate $\alpha$, so as to extrapolate the wind speed at 80 m. In mountainous

regions, the heights at 850 hPa, or 500 hPa may be closer to 80 m than 10 m above the surface; in that case, we use data at the next available level of 850 hPa or 500 hPa to derive the heights of that level and thus to extrapolate the wind speed at 80 m.

The horizontal resolution of the MERRA-2 is 0.5° in latitude (about 56 km) and 0.625° in longitude (about 53 km). The MERRA-2 reanalysis interpolates the data and the metadata at the exact output latitude

and longitude, hence the wind speed, air density and elevation refer to the grid points with the particular sets of latitude and longitude (Bosilovich et al., 2016). Hence, the longest distance between a wind farm and the its closest MERRA-2 grid-cell centre is about 39 km.

For energy-production data, we use the net monthly energy production of wind farms in Megawatt-hours (MWh) from the Energy Information Administration (EIA) between 2003 and 2016. Each of the

wind farms has a unique EIA identification number. After we neglect about 300 wind sites with incomplete or substantial zero production data, a total of 607 wind farms in the CONUS are selected in this analysis. For simplicity, the CONUS in this analysis is defined as the area bounded by 127°W, 65°W, 24°N and 50°N, and geographically includes the 48 states in CONUS and Washington, D.C. (Fig. 1).

### 2.2 Methodology

**2.2.1 Linear Regression and Data Post-Processing**

We focus on the direct relationship between wind speed and energy production to investigate approaches for calculating long-term variability. Therefore, we must minimize the influence from other determinants of energy production, such as curtailment and maintenance. First, we eliminate data with zero values for monthly energy productions, which is typical in the first months of a new wind farm. Next,

we linearly regress the monthly total energy production on the monthly mean MERRA-2 80-m wind speed at the closest grid point to each wind farm from 2003 to 2016. In other words, each wind site is assigned its own regression equation. We then remove any production data below the 90% prediction interval to exclude under-productions for reasons other than low wind speeds, and omit the data above the 99%



prediction interval, or potentially erroneous over-productions. Prediction intervals are calculated via the

t-values and the standard error of prediction (Montgomery and Runger, 2014).

After regressing the outlier-free energy data on wind speed, we then filter the wind farms based on the coefficient of determination ($R^2$), which indicates the confidence of the linear regression. We select the $R^2$ threshold of 0.75: 349 of the original 607 wind farms pass this filter. Considering some farms lack years of complete generation data, we extend the monthly energy production to 37 years using the same

site-specific linear models with the monthly MERRA-2 wind speed. In other words, we compute any missing energy-production data from 1980 to 2016 based on the linear fit from the years that do exist in the dataset. Herein, we refer this long-term extension of data as the predicted energy production. Of the 349 wind farms, 204 locations require seven or more years of derived energy data given the available EIA records between 2003 and 2016.

We then further apply a second filter using the Pearson's correlation coefficient ($r$) between the predicted and actual monthly energy productions, and only choose the 195 wind farms with $r$ larger than 0.8. As a result, of the r-filtered wind sites, we ensure wind speed is the primary driver of wind-power production, and we confirm the energy predictions match well with those observed.

The non-filtered, $R^2$-filtered and r-filtered wind farms carpet most of the popular wind-farm regions

across the CONUS (Fig. 1), even with the high $r$ threshold of 0.8. Thus, the r-filtered samples provide a sufficient representation of the wind farms across the United States. To illustrate our analysis with examples, we select one site in Oregon (OR) and another site in Texas (TX) that demonstrate distinct wind-speed distributions. We choose the two sites to contrast the results of different variability metrics throughout the paper; both sites pass the $r$ filter (Fig. 1).






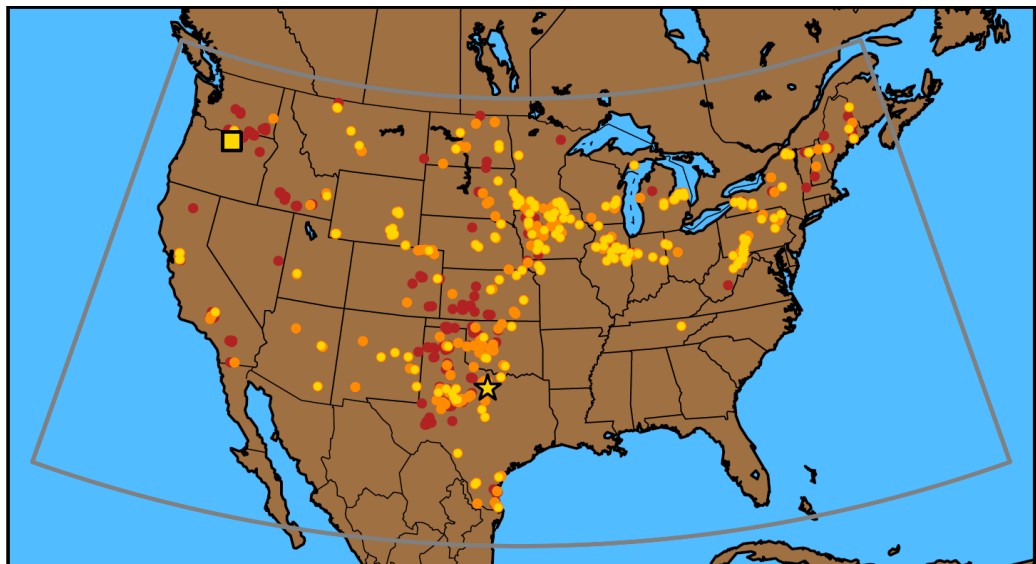

**Figure 1: Wind farm locations in the CONUS: non-filtered 607 sites in dark red, $R^2$-filtered 349 sites in orange, and r-filtered 195 sites in yellow. The yellow square represents the Oregon site and the yellow star indicates the Texas site (Table 2). The grey box illustrates the boundary of the CONUS used in this study.**


Recognizing that the horizontal resolution of the MERRA-2 data could be perceived as undermining the linear regressions, we explore any possible role of the distance between the closest MERRA-2 gird point and the actual wind farm, but we find no statistical relationship. In particular, horizontal and vertical discrepancies between the model and the observations do not affect the resultant $R^2$ in the linear

regressions. More than half of the 607 wind farms pass the $R^2$ filter, and more than half of those pass the $r$ filter (Fig. 2a). The distribution shapes of the horizontal distances and the elevation differences between the closest MERRA-2 gird point and the actual wind farm remain similar with the two filters applied (Fig. 2b and c). In other words, the horizontal and vertical distances between the MERRA-2 grid points and the wind farms have no apparent impact on the representativeness of the wind farms in the linear

regression.





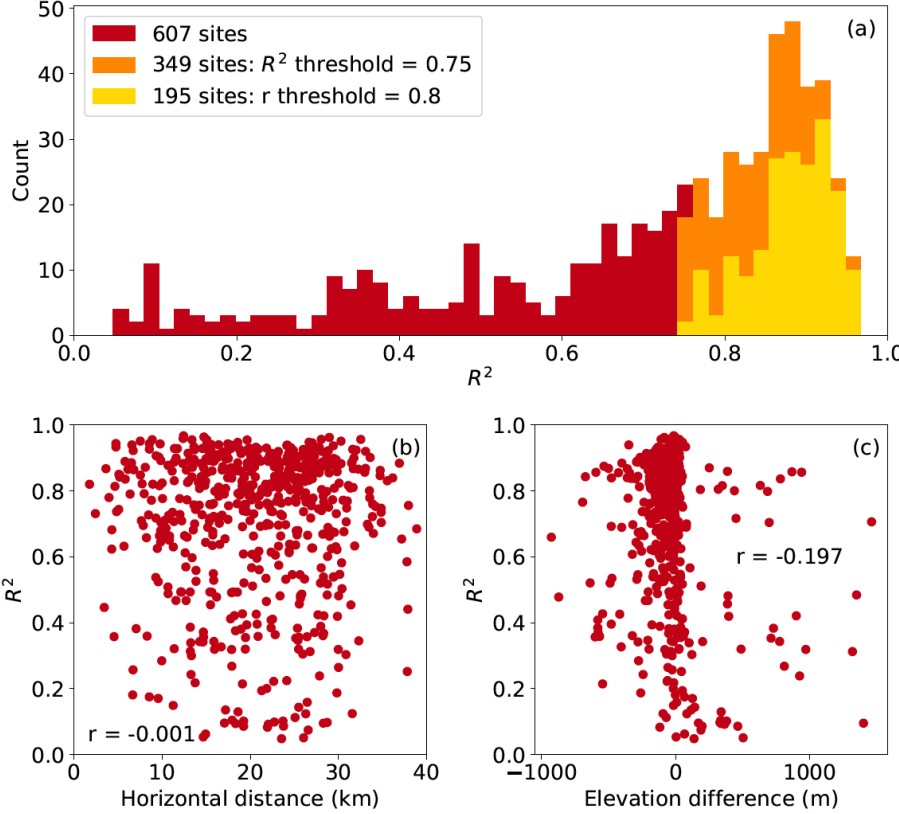

**Figure 2: (a)** Histogram of R² of all non-filtered sites (dark red), R²-filtered sites (orange) and r-filtered sites (yellow); **(b)** Scatterplot of the R² of all non-filtered sites and the horizontal distance between the closest MERRA-2 grid cell and the actual locations of the sites; **(c)** Scatterplot of the R² of all non-filtered sites and the absolute elevation difference between the closest MERRA-2 grid cell and the actual locations of the wind sites. The r in (b) and (c) represents the Pearson's correlation coefficient.

Additionally, we analyse the uncertainty of the linear regression method. We first test the influence of the error term in the regression, to account for the uncertainty associated with the input data. After a wind farm passes the $R^2$ threshold of 0.75, we add a random value within one standard error to the predicted energy production of each month. This random error term introduces uncertainty to the regression process but does not affect the $R^2$ of the site-specific regression. Furthermore, we also test the sensitivity of the $R^2$ and $r$ thresholds by analysing the results after modifying those limits. Specifically, we loosen the $R^2$ and $r$ thresholds to 0.6 and 0.7, and we tighten the $R^2$ and $r$ thresholds to 0.85 and 0.9.



Loosening these thresholds increases the sample sizes of the wind farms that pass the filters, and tightening the thresholds results in the opposite.

We test other factors that could undermine these regressions. We considered the hub-height air density extrapolated from MERRA-2 as another regressor in the regressions, but air density is a statistically insignificant predictor and thus is not discussed in the rest of this study. When we replace prediction

interval with confidence interval, the sample sizes increase from 349 and 195 sites to 555 and 209 wind farms. However, at least seven years of energy data are derived from the regression for 99% of the samples, because confidence intervals are smaller than prediction intervals by definition. We also considered removing the long-term means and the impacts of annual cycles, yet the sample sizes decrease to 121 and 69 locations, and the regression fills most of the energy data for over 99% of the sites. Finally,

to ensure these results were not specific to the MERRA-2 dataset, we perform the same analysis on the ERA-Interim reanalysis dataset (Dee et al., 2011). The results of the key variability parameters such as σ, CoV and RCoV resemble the findings using MERRA-2, hence we focus on the MERRA-2 findings in this study.

Our analysis, although comprehensive, is constrained by the quality of our data. On one hand,

reanalysis datasets have errors and biases in wind-speed predictions from complexities in elevation and surface roughness (Rose and Apt, 2016). Reanalysis datasets also demonstrate long-term trends of surface wind speeds as well (Torralba et al., 2017). The MERRA-2 dataset can also depict different meteorological environments than those at the wind-farm locations, especially in complex terrain. Thus, regressing actual energy production on reanalysis wind speed adds uncertainty to our analysis. On the

other hand, constrained by the monthly total energy-production data from the EIA, our analysis ignores the signals finer than monthly cycles. The quality of the EIA data also varies across wind sites, therefore the filtering process via linear regression is necessary.

**2.2.2 Variability Metrics Relating Wind Speeds and Energy Production**

To evaluate the variabilities of both the wind speeds and the predicted energy generations from the

filtered wind farms, we investigate a total of 27 combinations and variations of existing methods describing the spread of data. We categorize different variability metrics according to statistical



robustness (insensitivity to assumptions about the data, for instance, Gaussian distribution) and statistical resistance (insensitivity to outliers) (Wilks, 2011). Of the 27 variability methods tested, we select four representative measures to inter-compare and discuss in detail, according to their robustness, resistance,
and the nature of normalization by an average metric:

- RCoV, defined as the MAD divided by the median (Gunturu and Schlosser, 2012; Watson, 2014), is a spread metric divided by an average metric, and is both statistically robust and resistant;

- Range (maximum subtract minimum) divided by trimean (weighted average among quartiles) is a spread metric normalized by an average metric, and the numerator is not resistant;

- CoV (Baker et al., 1990; Bodini et al., 2016; Hdidouan and Staffell, 2017; Krakauer and Cohan, 2017; Rose and Apt, 2015; Wan, 2004), defined as the $\sigma$ divided by the mean, is a spread metric normalized by an average metric, and neither the denominator nor the numerator are robust or resistant;

- $\sigma$ is simply a spread metric that is not robust or resistant.

Among the four measures, only RCoV is completely statistically robust and resistant, and the first three
methods are all normalized spread metrics. We further describe all the tested variability methods comprehensively in Table B1. Each of these metrics is easy to implement via basic Python packages such as NumPy and SciPy with no more than a few lines of code. In addition, based on the exponential scaling relationship between power and wind speed developed by Bandi and Apt (2016), we also analyse the results from the exponential CoV and the exponential RCoV in this paper (Table B1).

230        In addition to calculating variabilities with the spread measures, we evaluate other diagnostics that describe distribution characteristics. These diagnostics include averaging metrics such as the arithmetic mean (not resistant) and median (the 50th percentile, which is resistant), symmetry metrics such as skewness (involving the third moment, not robust or resistant) and Yule-Kendall Index (YKI, robust and resistant), a tailedness metric, namely kurtosis (involving the fourth moment, not robust or resistant), the
Weibull scale and shape parameters (not robust), and the autocorrelation with 1-year lag to dissect the inter-annual cycles. We summarize the diagnostics evaluated in this analysis in Table B2. Along with the regression results, results from the four representative variability metrics and other distribution diagnostics demonstrate differences between the two selected sites (Table 2).





Herein, we quantify the variabilities of the 37-year extended time series of wind speed and energy
production via different methods, using a range of time frames: 1 year, 2 years, and up to 37 years for
each wind farm. A metric is considered useful when the resultant wind-speed variability correlates well
with the resultant energy-production variability across wind farms, even when random errors are
implemented and the thresholds $R^2$ and $r$ are changed. In this analysis, we inter-compare results with
three correlation metrics: Pearson's $r$, Spearman's rank correlation coefficient ($r_s$) and Kendall's rank
correlation coefficient ($\tau$) (Table 1).

**Table 1: Details of the three correlation metrics applied, adapted from Wilks (2011). All three metrics yield values between -1 and 1.**

| Correlation metrics | Robust and resistant | Description |
|---|---|---|
| Pearson correlation coefficient ($r$) | No | Calculate the covariance of x and y, divided by the product of standard deviations of x and y |
| Spearman's rho, or Spearman rank correlation coefficient ($r_s$) | Yes | Transform x and y values into ranks within x and y themselves, then calculate the covariance of ranks in x and y, divided by the product of standard deviations of ranks in x and y |
| Kendall's tau, or Kendall's rank correlation coefficient ($\tau$) | Yes | Match all data pairs between x and y, with $\frac{n(n-1)}{2}$ matches possible with sample size of $n$. Define concordant pair as both $x_1$ larger than $x_2$ and $y_1$ larger than $y_2$, or both $x_1$ smaller than $x_2$ and $y_1$ smaller than $y_2$. Define discordant pair as either $x_1$ larger than $x_2$ and $y_1$ smaller than $y_2$, or $x_1$ smaller than $x_2$ and $y_1$ larger than $y_2$. Calculate $\tau = \frac{2(Concordant\ pairs - Discordant\ pairs)}{n(n-1)}$ |


To assess the applicable time frames of various variability metrics, we evaluate the asymptote period
of correlations for each method. In most cases, the correlation coefficients asymptote to the 37-year value
after a certain analysis time frame. Using RCoV as an example, the Pearson's $r$'s of shorter analysis
periods (1-year, 2-year, etc.) gradually converge to the 37-year value at 0.856 as the RCoV-calculation
time frame expands (Fig. 5a). Hence, for each metric, assuming the 37-year correlation coefficient



represents the long-term correlation, we calculate the normalized differences between the correlation coefficients and the 37-year value in each time frame, starting from 1-year. When the absolute mean of the normalized differences drops below 0.05 in a particular year, we determine that year as the length of data required for reliable results via that variability method. In other words, the asymptote year of a certain

metric illustrates that the error of the resultant correlation between wind-speed and energy-production variability via that data length is under 5% from the long-term value. For example, the asymptote period of RCoV correlations is 3 years according to Pearson's $r$ (Table 3).

To relate the IAVs between wind speed and energy production, we also perform the same analysis for annual-mean data. Strictly speaking, calculating the variabilities using monthly mean data yield inter-

monthly variabilities, because the results account for monthly, seasonal and annual signals. To isolate the signals from inter-annual variations, we also examine the metrics and their correlations between the annual means of hub-height wind speeds and energy productions, after linear regressing and filtering via monthly data. However, the sample of each site are then limited to 37 data points of annual wind speed and energy production. Besides, selecting de-trend data from long-term means to calculate variabilities

and their correlations leads to trivial results because of the small sample sizes, and hence is omitted in this study.

### 2.2.3 Investigation of Wind-Speed RCoV

After we demonstrate RCoV is the most systematic approach in linking wind-speed and energy-generation variabilities in Section 3.2, we further examine the details of using RCoV, specifically

determining the minimum length of wind-speed data necessary to quantify variability effectively. We use 37 years of wind speed in every MERRA-2 grid cell in the CONUS (a total of 5049 grid points), and we calculate the RCoVs with 1 to 37 years of data for each grid cell. Because the RCoVs calculated using data between 1980 to 2016 are only samples of the true long-term wind-speed variability and hence the results involve uncertainty, we select a confidence interval approach.

We assume that the distribution of RCoV is Gaussian with infinite years of wind speed. Hence, we use a chi-square ($\chi^2$) distribution to set bounds for the standard deviations from samples of RCoV. In other words, because the derived RCoVs differ with years of wind speeds sampled, we use the $\chi^2$




distribution to quantify the confidence intervals of RCoV for each sample size. To determine the
minimum data required for RCoV calculation, we use the following criterion (Montgomery and Runger,
285   2014):

$$\sigma_{37} \geq |\sqrt{\frac{(n_i-1)\sigma_i^2}{\chi_{\alpha/2,n_i-1}^2}}| \, , \tag{3}$$

where $\sigma_{37}$ is the per-determined 37-year σ of RCoV, $n_i$ is the sample size of n years in year i which is
between 1 to 36 years, $\sigma_i^2$ is the variance of the sample of RCoVs in year i, and $\chi_{\alpha/2,n_i-1}^2$ is the percentage
point of the $\chi^2$ distribution given the confidence level of $\alpha$ and the degrees of freedom of $n_i - 1$. We
select a pair of $\alpha$ levels, 90% and 95%, hence we use four percentage points of the $\chi^2$ distribution at
0.025, 0.05, 0.95 and 0.975 to construct the respective confidence intervals. Because the 37-year RCoV
is an estimate of the truth, which is the wind-speed RCoV of infinite years, its singular value does not
yield any variance or possess any distribution shape. Thus, to construct the confidence interval of the
standard deviation of the truth, we set the pre-determined $\sigma_{37}$ as a fraction of the 37-year RCoV.
Particularly, the $\sigma_{37}$ are 10% and 5% of 37-year RCoV for the 90% and 95% confidence levels
respectively.

In summary, for each grid point, we first determine an uncertainty bound based on the 37-year wind-
speed RCoV of the location: we assign a 37-year $\sigma$, which is either 5% or 10% of the 37-year RCoV,
dependent on the confidence level, either 95% or 90% confidence. For each year $i$, from 1 to 37 years,
we calculate the pairs of $\chi^2$-derived $\sigma$'s of year $i$, which represent the lower and upper bounds of the
confidence interval. When both of the $\chi^2$-derived $\sigma$'s become smaller than the per-determined 37-year $\sigma$,
year $i$ becomes the minimum length of data required to calculate RCoV effectively at the specific
confidence level. We analyse the wind-speed RCoV via both monthly mean and annual-mean wind
speeds. We label the resultant minimum length of wind-speed data based on the $\chi^2$ method as
convergence year, in contrast to the asymptote period which determines the asymptote year of correlation
coefficients.



## 3 Results

### 3.1 Case Studies: OR and TX sites

We select two sites from two different geographical regions with considerable wind-energy
deployment, the southern Plains and the Pacific Northwest in the United States, to contrast the results of
various variability metrics. Based on the site-specific regressions, we extend the monthly energy-
production time series to 37 years (Fig. 3a and b) for the two sites. Both sites pass the $R^2$-filter at 0.75
and the $r$-filter at 0.8. Although the OR site is farther from the closest MERRA-2 grid point in a region
with more complex terrain, the resultant $R^2$ (0.87) and predicted-actual energy Pearson's $r$ (0.91) are
larger than those of the TX site (0.79 and 0.81 respectively) (Table 2). The 37-year-average wind speed
of about 7.6 m s$^{-1}$ at the TX site is larger than that of the OR site at about 6.8 m s$^{-1}$ (Table 2). Additionally,
the 12-month-lag autocorrelations demonstrate that the annual cycle of monthly wind speeds of the TX
site is stronger than that of the OR site, yet the autocorrelations of the sites, 0.53 and 0.32, are still lower
than the CONUS median of 0.58 (Table 2).






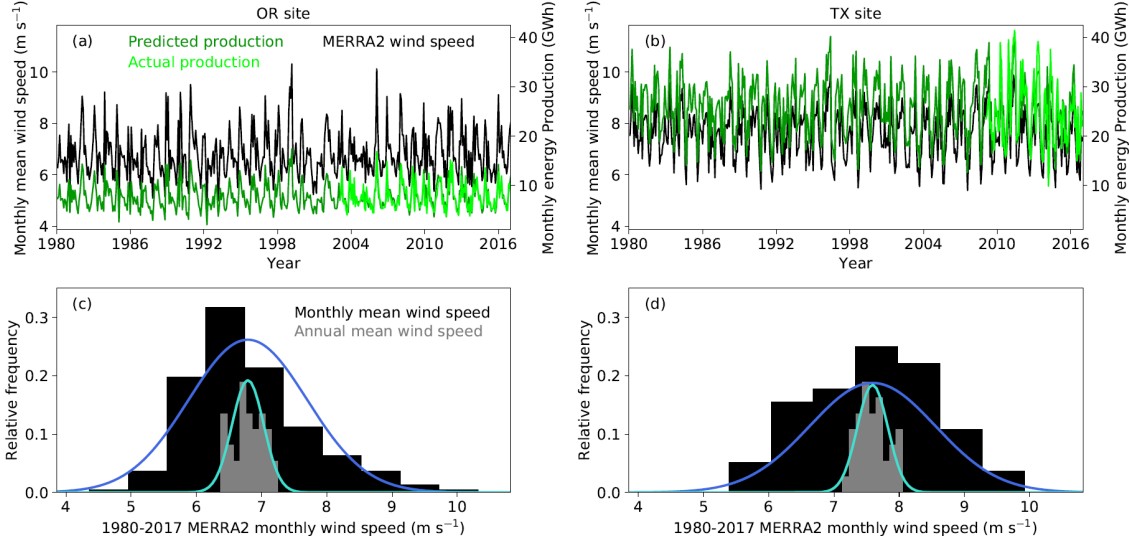

**Figure 3: (a) Time series of MERRA-2 monthly mean 80-m wind speed (black), actual monthly net EIA energy production (lime), and extended monthly energy production from 1980 to 2016 based on linear regression (green) at the OR site; (b) Time series at the TX site with the same annotations as in (a); (c) Histograms of MERRA-2 monthly mean wind-speed distribution (black) and yearly-mean wind-speed distribution (grey) at the OR site from 1980 to 2016. The blue curve indicates the Gaussian fit of the monthly mean wind speeds via the mean and the σ, and the cyan curve represents the Gaussian fit of the annual-mean data; (d) Histograms and curves of Gaussian fit of wind-speed distributions at the TX site with the same annotations as in (c).**






**Table 2: Site details, monthly means, and annual means of various metrics at the two selected sites based on 37 years of monthly and annual wind speeds, 37 years of predicted energy productions, and actual energy productions; and the CONUS medians of wind-speed metrics using 37 years of monthly and annual mean data.**

| Site specifics | OR site | | TX site | | CONUS median | |
|---|---|---|---|---|---|---|
| Location, region and state | Condon, Columbia Gorge, OR | | Bryson, northwest of Fort Worth, TX | | 5049 MERRA-2 grid points | |
| Nominal capacity (MW) | 24.6 | | 120 | | / | |
| Elevation at closest MERRA-2 grid point - elevation of actual wind farm (m) | -501.4 | | -67.4 | | / | |
| Horizontal distance between MERRA-2 location and actual location (km) | 33.07 | | 21.22 | | / | |
| $R^2$ of final linear regression | 0.868 | | 0.794 | | / | |
| RMSE of final linear regression (MWh) | 1140.5 | | 4185.0 | | / | |
| Pearson's $r$ between predicted and actual energy | 0.906 | | 0.809 | | / | |
| Variability metrics | Monthly mean | Annual mean | Monthly mean | Annual mean | Monthly mean | Annual mean |
| 37-year wind-speed RCoV | 0.082 | 0.029 | 0.094 | 0.023 | 0.102 | 0.021 |
| 37-year energy-production RCoV | 0.226 | 0.059 | 0.166 | 0.041 | / | / |
| Actual energy-production RCoV | 0.233 | 0.067 | 0.212 | 0.055 | / | / |
| 37-year wind-speed $\frac{range}{trimean}$ | 0.893 | 0.129 | 0.596 | 0.122 | 2.066 | 1.316 |
| 37-year energy-production $\frac{range}{trimean}$ | 2.050 | 0.288 | 1.059 | 0.218 | / | / |
| Actual energy-production $\frac{range}{trimean}$ | 1.768 | 0.307 | 1.303 | 0.305 | / | / |
| 37-year wind-speed CoV | 0.134 | 0.036 | 0.127 | 0.031 | 0.143 | 0.031 |
| 37-year Energy-production CoV | 0.333 | 0.081 | 0.225 | 0.055 | / | / |
| Actual energy-production CoV | 0.341 | 0.088 | 0.279 | 0.089 | / | / |
| 37-year wind-speed σ | 0.909 | 0.242 | 0.964 | 0.234 | 0.895 | 0.203 |
| 37-year energy production σ | 2.599 | 0.632 | 5.828 | 1.421 | / | / |
| Actual energy-production σ | 2.663 | 0.687 | 6.964 | 2.228 | / | / |
| Other 37-year wind-speed diagnostics | Monthly mean | Annual mean | Monthly mean | Annual mean | Monthly mean | Annual mean |
| mean (m s$^{-1}$) | 6.79 | 6.79 | 7.59 | 7.59 | 6.45 | 6.45 |
| median (m s$^{-1}$) | 6.64 | 6.79 | 7.63 | 7.57 | 6.51 | 6.45 |
| kurtosis | 0.886 | -0.962 | -0.663 | -0.872 | -0.482 | -0.373 |
| skewness | 0.811 | -0.129 | -0.074 | 0.172 | 0.045 | 0.061 |
| YKI | 0.153 | 0.101 | -0.072 | 0.041 | -0.024 | 0.023 |
| 12-month-lag autocorrelation | 0.324 | 0.039 | 0.525 | -0.052 | 0.578 | 0.023 |




None of the monthly and annual wind-speed distributions of the sites are perfectly Gaussian.
According to the kurtosis, skewness and YKI values of the monthly-mean wind speeds (Table 2), the monthly wind-speed distribution at the OR site skews towards lower wind speeds with more and stronger extremes (Fig. 3c). The skewed distribution at the OR site leads to 71.2% of the monthly wind speeds locating within 1 $\sigma$ from the mean, compare to the classic Gaussian of 68.3%. Nevertheless, although the TX site monthly wind-speed distribution is very close to symmetric with fewer outliers (Fig. 3d), which
is supported by near-zero skewness and YKI (Table 2), only 64.6% of monthly data fall within 1 $\sigma$ from its mean. For annual-mean wind speeds, the averaging with a 12-month time span at both sites reduces the ranges, and thus leads to kurtosis close to -1 (Table 2). Although the skewness and YKI are close to 0 (Table 2), only 59.5% and 56.8% of the annual-mean wind speeds fall within 1 σ from the means of the OR and TX sites respectively.

The four selected variability methods yield similar resultant monthly variabilities that are close to the respective CONUS medians based on the 37-year monthly data. For variabilities of monthly wind speeds, the differences between the two sites are slight because the comparison among the results of the four metrics is inconclusive (Table 2): the monthly variabilities are not far from the national medians (Table 2). However, results from the normalized spread metrics (RCoVs, range divided by trimean, and CoV)
using the 37-year and the observed energy production illustrate that the OR site generates more variable wind power than the TX site (Table 2). The magnitudes of the variabilities between the 37-year and the actual monthly energy productions are also comparable, and the discrepancies between them are larger at the TX site than the OR site. Nonetheless, the predicted and the observed monthly energy productions of the two sites demonstrate similar variability characteristics overall.

Moreover, when we apply the four selected methods to the annual-mean data, the metrics describe IAV exactly. For both variables, wind speed and energy generation, nearly all metrics illustrate that the OR site has stronger IAV than the TX site, except for using $\sigma$ to quantify energy-production IAV (Table 2). Echoing the results of the monthly data above, using normalized metrics suggest the energy production at the OR site varies more than that at the TX site, inter-monthly and inter-annually. Note that all the
IAVs are smaller than the variabilities calculated using monthly data (Table 2), because the annual averaging collapses variations in the data.



Additionally, the magnitudes of energy variabilities and IAVs are also nearly or more than twice as large as those of wind speed (Table 2). The reason is the nature of the power curve: wind-power generation is a function of wind speed cubed at wind speeds below rated. Therefore, small wind-speed variations

propagate into large energy-production fluctuations that are discernible in monthly and yearly data.

**3.2 Variability Metrics Comparisons**

Matching the wind-speed and energy variabilities over 37 years at each $r$-filtered site, RCoV, as a statistically robust and resistant metric, yields the highest Pearson's $r$ (0.86) among the four highlighted methods as well as all the variability metrics evaluated (Fig. 4 and Table B1). A perfect variability

measure would link wind-speed and wind-power variations closely together with a correlation of unity, and so RCoV, with the highest Pearson's $r$, is the best of all. On one hand, a strong correlation between the wind-speed RCoV and the energy-production RCoV implies that the high wind-speed variability at a wind farm translates to high energy-generation variability, and vice versa (Fig. 4a). For instance, the moderate 37-year wind-speed RCoVs of the OR and TX sites indicate modest fluctuations in energy

productions between months (Fig. 4a). On the other hand, a non-resistant method, range divided by trimean, leads to a lower $r$ (0.64) and suggests the OR site has variable wind speed and energy production (Fig. 4b). For the other two non-robust and non-resistant methods, the CoV results in a modest $r$ (0.70) with a similar scatter as the RCoV (Fig. 4c); the $\sigma$, not normalized by an average metric, does not relate wind-speed and energy variabilities effectively (Fig. 4d). The positions of the two wind farms relative to

the rest of the sites in Fig. 4 illustrate that the TX site experiences average variabilities in wind resource and energy production, whereas the OR site has above-average energy-generation variability. Overall, the four methods lead to different representations of energy variability at the OR site.





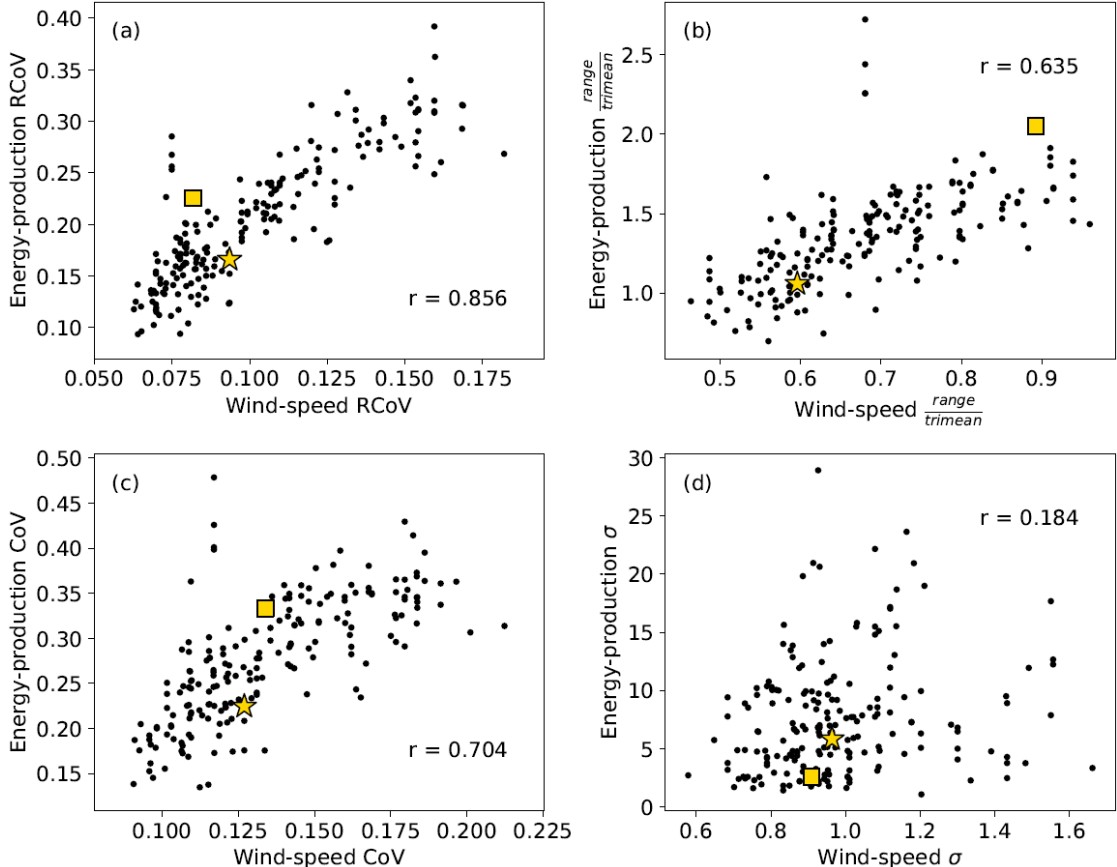

**Figure 4: Scatterplots of 37-year wind-speed variability and energy variability via four metrics: (a) RCoV, (b) $\frac{range}{trimean}$ , (c) CoV and (d) σ, based on monthly data from the 195 r-filtered wind sites. Each black dot represents each filtered site, and the $r$ value at the corner of each panel indicates the Pearson's $r$ between each pair of wind-speed and energy spread metrics. The yellow square and the yellow star denote the OR and the TX sites respectively.**

By increasing the years included in the variability calculations using monthly data, the resultant correlations of most metrics vary less, the correlations gradually converge to their 37-year values, and their asymptote periods vary. The 37-year Pearson's $r$ values from the four selected metrics between wind-speed and energy-production variabilities in Fig. 4 transform into the 37-year marks in Fig. 5, and we use a 5% threshold of normalized deviation to determine the asymptote periods. Particularly, the $r$'s




from RCoV and CoV (Fig. 5 a and c) reach their respective asymptotes steadily with longer length of data, whereas the $r$'s from range divided by trimean does not (Fig. 5b). The 37-year correlation using $\sigma$ is weak and thus the method is not actually useful: while the $r$'s approach the 37-year benchmark (Fig. 5d), this correlation value is so low (0.2) as to be not effective. Paired with a high long-term $r$, the asymptote period of a metric indicates the appropriate time span of wind-speed data required to represent

the variability of wind-energy production. For example, the resultant $r$'s using RCoV asymptote to a high value after just 3 years, meaning one needs 3 years of wind-speed data to estimate the wind-speed variability so as to adequately infer the energy-production variability of a certain or potential wind farm via RCoV.

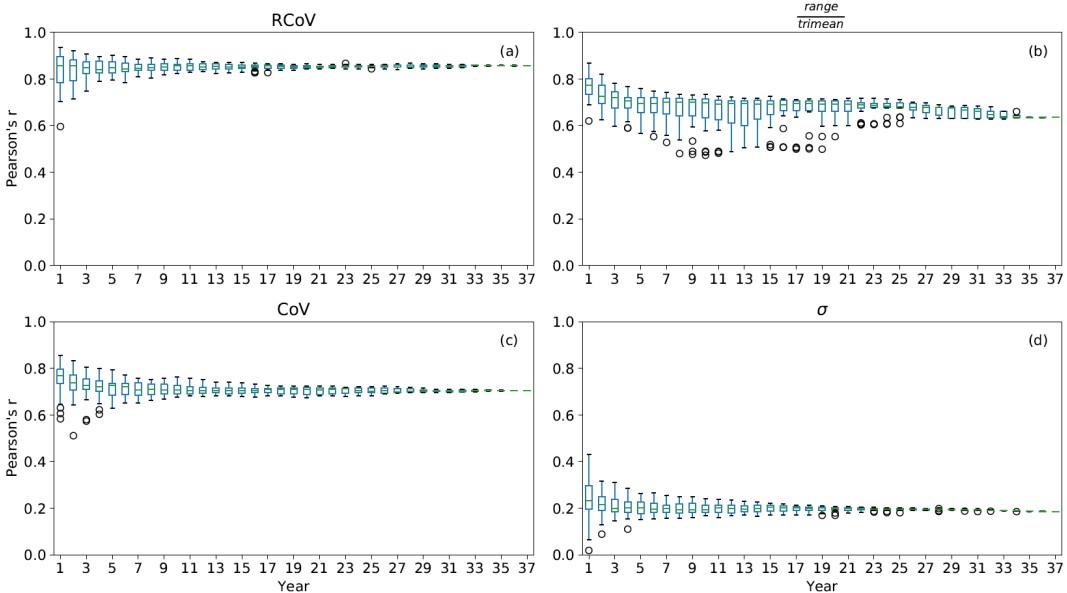


**Figure 5: Boxplots of Pearson's $r$ between wind-speed variability and energy variability for differet analysis time frames, from 1 year to 37 years: (a) RCoV, (b) $\frac{range}{trimean}$ , (c) CoV and (d) σ, based on the monthly data from the 195 r-filtered wind sites. Each $r$ represents the correlation using all the filtered sites of a particular time frame. The 37-year correlations equal to the $r$ values listed in Fig. 4. The box and whiskers represent the third quartile plus the 1.5 times of interquartile range (IQR), the third quartile, the**

**median, the first quartile, and the first quartile minus the 1.5 times of IQR.**



The three correlation coefficients (Pearson's $r$, Spearman's $r_s$, and Kendall's $\tau$) yield consistent results among all variability metrics tested, hence we primarily present the results using Pearson's $r$ herein. Table 3 summarizes the 37-year correlations ($r$, $r_s$ and $\tau$), between the wind-speed variabilities

and the energy-production variabilities using the $r$-filtered data, and the respective asymptote periods of the methods. The $r$ and $\tau$ of RCoV are the largest (0.86 and 0.67 respectively) among all variability metrics, and the associate asymptote periods are also relatively short (2 to 3 years) (Table 3). Another normalized, robust, and resistant spread metric, interquartile range (IQR) divided by median, results in the highest $r_s$, and the $r_s$ of RCoV is the second largest (Table 3). More importantly, the asymptote periods

of RCoV are the smallest of all, regardless of the choice of correlation coefficient. In other words, fewer years of data are necessary to calculate RCoV to effectively relate wind-speed and energy variabilities than any other metric. Overall, when a spread metric yields strong correlations between variabilities of wind speed and energy generation, the correlation metrics agree with each other (Table 3). Therefore, the results in this paper focus on Pearson's $r$, which is a commonly used correlation coefficient.

In addition to the spread metrics, other distribution diagnostics also yield strong correlations between the 37-year monthly wind speed and energy production. For example, kurtosis and skewness result in $r$ and $r_s$ above 0.9. Since we determine the asymptote periods based on normalized deviations, when the 37-year correlation benchmark of a metric is high, the respective asymptote period tends to be shorter. Therefore, only 1 year of monthly data is required to compute kurtosis and skewness adequately, except

for using $r_s$ in kurtosis, where those $r_s$'s of smaller number of years are low. (Table 3). Moreover, the symmetry and the shape of energy-production distribution can be characterized using wind-speed data, given the moderately strong correlations of YKI and Weibull shape parameter (Table 3).



**Table 3: Correlations and the associated asymptote periods of wind-speed variability and energy variability using various methods,**
**diagnositcs with different correlation metrics, based on the monthly data of the 195 r-filtered wind sites.**

| Spread metrics | 37-year $r$ | Asymptote years from $r$ | 37-year $r_s$ | Asymptote years from $r_s$ | 37-year $\tau$ | Asymptote years from $\tau$ |
|---|---|---|---|---|---|---|
| CoV | 0.704 | 5 | 0.754 | 4 | 0.565 | 9 |
| $\dfrac{\sigma}{median}$ | 0.743 | 4 | 0.781 | 3 | 0.595 | 4 |
| $\dfrac{\sigma}{trimean}$ | 0.728 | 4 | 0.770 | 3 | 0.583 | 6 |
| $\dfrac{IQR}{mean}$ | 0.818 | 4 | 0.821 | 3 | 0.636 | 6 |
| $\dfrac{IQR}{median}$ | 0.845 | 3 | 0.843 | 3 | 0.662 | 6 |
| $\dfrac{IQR}{trimean}$ | 0.834 | 3 | 0.834 | 3 | 0.650 | 6 |
| RCoV | 0.856 | 3 | 0.836 | 2 | 0.663 | 3 |
| $\dfrac{MAD}{mean}$ | 0.834 | 3 | 0.822 | 3 | 0.648 | 5 |
| $\dfrac{MAD}{trimean}$ | 0.848 | 3 | 0.832 | 3 | 0.660 | 5 |
| $\dfrac{Range}{mean}$ | 0.609 | 30 | 0.711 | 28 | 0.516 | 31 |
| $\dfrac{Trimmed\ \sigma}{median}$ | 0.806 | 3 | 0.807 | 3 | 0.631 | 5 |
| $\dfrac{Trimmed\ \sigma}{trimean}$ | 0.794 | 4 | 0.801 | 4 | 0.622 | 6 |
| Seasonality Index, modified from Walsh and Lawler (1981) | 0.744 | 5 | 0.766 | 4 | 0.584 | 7 |
| Other diagnostics | | | | | | |
| Kurtosis | 0.936 | 1 | 0.934 | 14 | 0.785 | 24 |
| Skewness | 0.943 | 1 | 0.938 | 1 | 0.798 | 18 |
| YKI | 0.778 | 23 | 0.712 | 33 | 0.538 | 34 |
| Weibull shape parameter | 0.721 | 4 | 0.741 | 5 | 0.559 | 7 |





Additionally, we also perform the same correlation and asymptote analyses on the data from changing
the $R^2$ and $r$ filter thresholds as well as the data with random error, and RCoV again yields the strongest
correlations and the shortest asymptote periods among all methods. We adjust the $R^2$ and $r$ requirements
in the linear-regression process, thus changing the filtered sample sizes. On one hand, reducing the $R^2$
threshold to 0.6 and $r$ threshold to 0.7 increases the respective sample sizes to 461 and 306 wind farms,
but weakens the correlations between wind-speed and energy variabilities for all methods (Table B3). On
the other hand, increasing $R^2$ threshold to 0.85 and $r$ threshold to 0.9 strengthens the wind speed-energy
correlations of all the metrics, and shrinks the sample sizes to 212 and 83 wind farms respectively (Table
B3). Modifying the filtering thresholds leads to different $r$'s yet similar asymptote periods among all
metrics. Moreover, we also test the vigorousness of our findings by introducing an error term, randomized
based on the standard error, in predicting the 37-year energy productions. The error term adds uncertainty
to resemble the reality of noisy wind-speed and power-production data. We introduce the error term to
the predicted energy productions for each of the 349 wind farms that pass the original $R^2$-threshold of
0.75. This approach weakens the correlations and lengthens the asymptote periods for most metrics (Table
B3). Overall, according to the results from the $R^2$-r-threshold and the random error tests, RCoV yields the
highest $r$'s among all methods, and its asymptote periods remain reasonably short.

Further, normalized and simple spread metrics yield different relative wind-speed variabilities
between wind sites. On one hand, the correlations coefficients between 37-year monthly mean wind-speed
RCoV and CoV, two spread metrics that are normalized by average metrics, are nearly unity (Fig. 6a).
The comparison between two simple spread metrics, MAD and σ, result in correlation coefficients close
to 1 also (Fig. 6d). The relative positions of OR site highlight the differences between Fig. 6a and Fig.
6d: compare to other wind farms, the OR site has moderate wind-speed RCoV and CoV, but small MAD
and $\sigma$. Compared to Fig. 6a, the lower $r_s$ and $\tau$ in Fig. 6d illustrate that MAD and $\sigma$ can misrepresent the
relative wind-speed variabilities of a wind site. On the other hand, the results between a normalized spread
metric (RCoV and CoV) and the respective simple spread metric (MAD and $\sigma$), which is also the
numerator of the normalized spread metric, lead to weaker correlations (Fig. 6b and c). The $r$, $r_s$ and $\tau$
between 37-year monthly wind-speed RCoV and $\sigma$ are 0.684, 0.738, and 0.579 respectively (not shown).
The wind sites with slower average wind speeds and thus disproportionately larger normalize spread




results cause the deviations from perfect correlations in Fig. 6b and c. Therefore, normalized spread
metrics, which account for the differences in wind-speed magnitude, become advantageous over simple
spread metrics in comparing variabilities of wind sites. Note that we demonstrate similar comparisons
between wind-speed spread metrics via annual-mean data in Fig. A2.

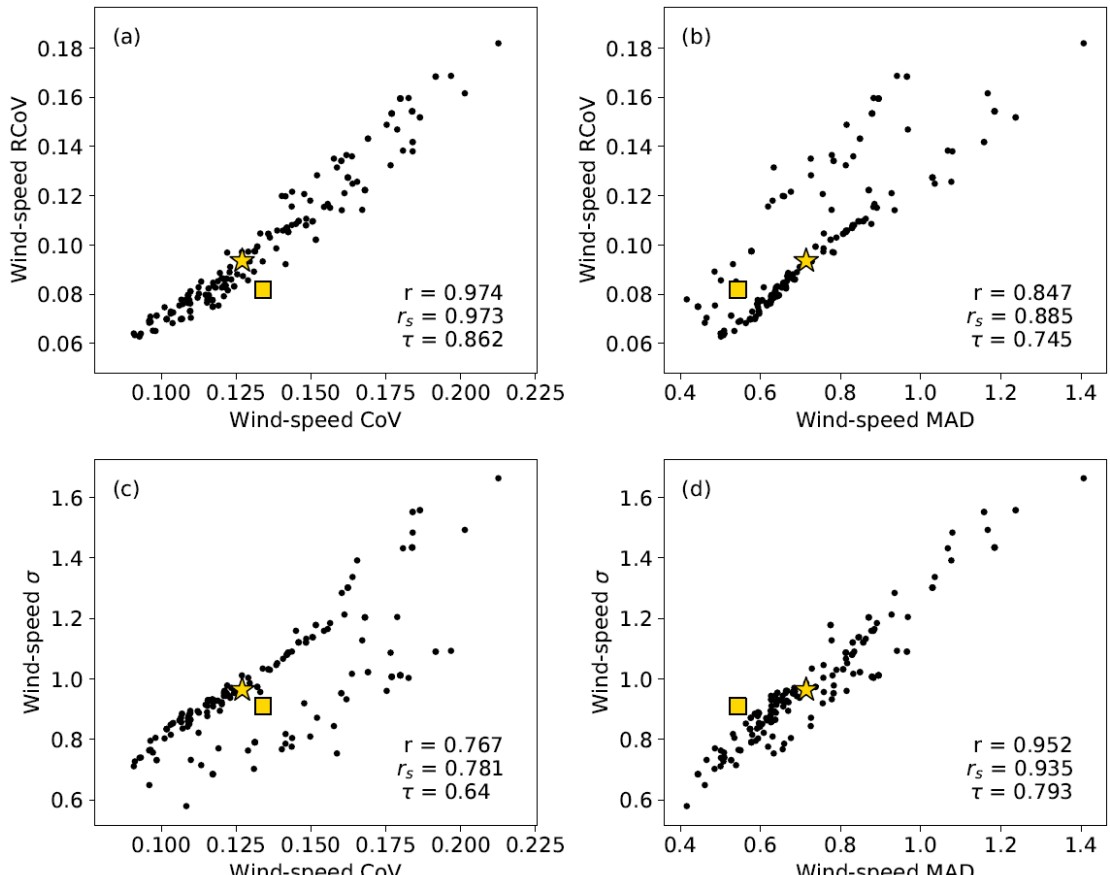


**Figure 6: Similar to Fig. 4, but for scatterplots to compare 37-year wind-speed variability metrics: (a) RCoV and CoV, (b) RCoV and MAD, (c) σ and CoV, and (d) σ and MAD, based on monthly data from the 195 r-filtered wind sites. Each black dot represents each filtered site, and the $r$, $r_s$ and $\tau$ at the corner of each panel indicate the Pearson's $r$, the Spearman's rank correlation coefficient and the Kendall's rank correlation coefficient between each pair of wind-speed spread metrics. The yellow square and the yellow**

**star denote the OR and the TX sites respectively.**



Meanwhile, using annual-mean data to compute IAVs can lead to misleading interpretations. Scatterplots of the 37-year wind-speed and energy IAVs similar to Fig. 4 are illustrated in Fig. A1, via the same 195 $r$-filtered sites. The correlations via yearly averages are generally weaker except for a few

metrics, including range divided by mean which yields the largest $r$ of all (Table B4). However, the 37-year correlations do not adequately represent the long-term values (Table B4), so even the resultant asymptote periods are longer than those using monthly data, the asymptote analysis method is unsuitable for annual data. Moreover, using annual averages greatly limits the sample size at each site even with 37 years of hourly wind-speed data. Statistically, a smaller sample leads to a smaller spread of that

distribution. Accordingly, with few years of data, small spreads in annual-mean wind speeds result in a tight cluster of IAVs among all the wind farms. Therefore, the compact collection of wind-speed and energy-production IAVs causes strong correlations, solely because of the small number of annual averages used in the IAV calculation. Thus, the correlations via annual means demonstrate a downward trend with increasing length of data, regardless of the variability metrics chosen (Fig. 7). Although the

correlations asymptote to the 37-year values, the weakening correlations with more years included in the IAV calculations imply that using less data is preferred in connecting the two IAVs. Note that the spread cannot be computed with one data point and hence the correlations between wind-speed IAVs and energy IAVs do not exist with a single year of data (Fig. 7). Overall, the asymptote analysis causes deceiving results, and given the nature of the annual data, we cannot determine the sufficient length of data to

effectively link the IAVs of wind speed and energy production. In other words, relating wind-speed IAV and energy-generation IAV with annual-mean data is flawed.




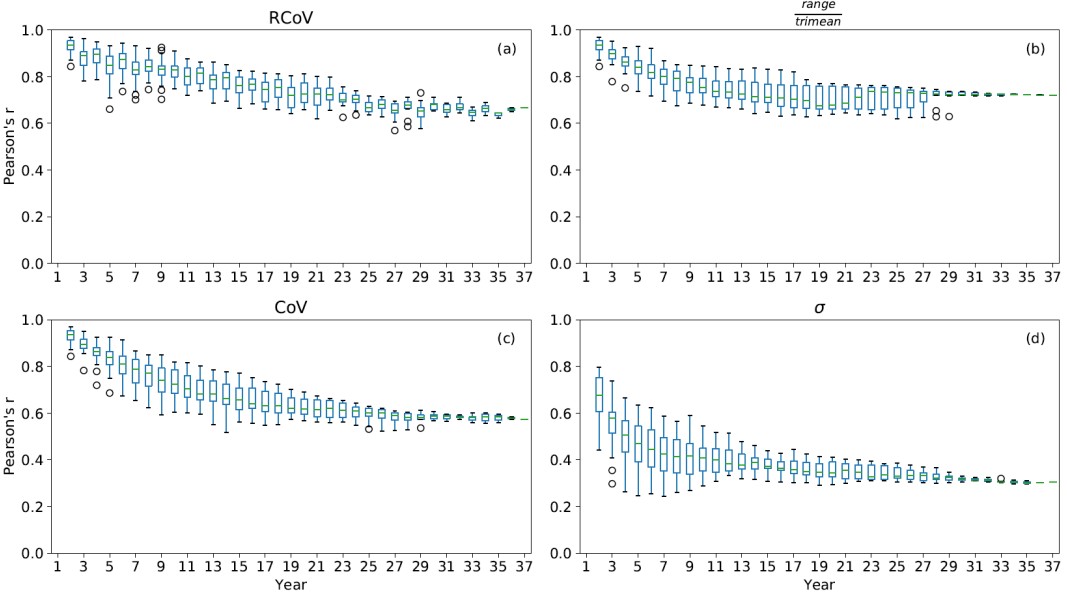

**Figure 7: As in Fig. 5, but for annual-mean data.**

### 3.3 Wind-speed RCoV Calculation and Spatial Distribution

Now that we have established that RCoV is a powerful and accurate way to relate wind-speed and energy-generation variations, we assess the required length of data to calculate the RCoV of wind speed. We compute the site-specific RCoVs using different spans of monthly mean wind speeds, including the OR and the TX sites (Fig. 8). The variations of RCoVs decrease with more years included in the calculations, and for each location we use the 37-year wind-speed RCoV as the long-term benchmark.

For example, the 37-year wind-speed RCoV of 0.082 at the OR site means that the median among the absolute deviations from the median is 8.2% of the median monthly mean wind speed (Fig. 8a and Table 2). We determine the 37-year $\sigma$'s as 10% and 5% of the 37-year RCoV, and we apply the $\chi^2$ approach at 90% and 95% confidence levels respectively to derive the convergence years, or the minimum length of wind-speed data required to calculate RCoV effectively. The convergence years of the OR and TX sites

are 12 and 25 years with 90% confidence, and 20 and 31 years with 95% confidence respectively (Table





B5). In other words, for the OR site, one needs 12 years of monthly mean wind speeds to compute RCoV
with 90% confidence that the resultant RCoV is within 10% deviation from the 37-year RCoV.

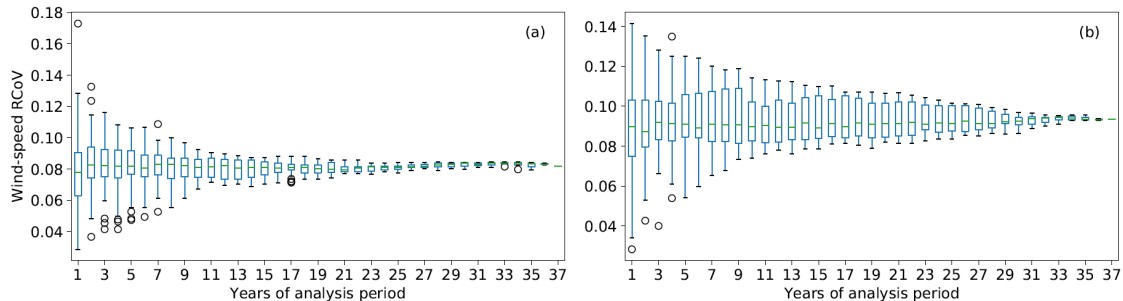

**Figure 8: Boxplots of wind-speed RCoV using monthly MERRA-2 data for different time frames from 1 year to 37 years at (a) the**
**OR site and (b) the TX site.**

To quantify the inter-monthly variability of wind speed at a wind farm, RCoV requires 10 years of
monthly wind-speed record with 90% confidence. In general, the $\sigma$'s of wind-speed RCoVs across the
CONUS decrease with more years included in the RCoV calculation (Fig. 9a). For each grid point, the
sample size of RCoV also becomes smaller, from 37 RCoVs of 1 year of data to 1 RCoV of 37 years of
data, and hence the $\sigma$ of RCoV decreases with the length of wind-speed records (Fig. 9a). With the $\sigma$'s
of RCoVs across 37 years, we determine the convergence years via the $\chi^2$ method. For a certain
confidence level, the cumulative fraction of the CONUS grid cells that exceed the associated threshold of
$\chi^2$-derived confidence intervals increases with the length of data (Fig. 9b). Among all of the MERRA-2
grid cells in the CONUS, the median convergence year is 10 years and the associated MAD is 3 years at
90% confidence level (Fig. 9b and Table B5). In other words, to assess the wind-speed variability via
RCoV with a maximum of 10% error from the long-term value and 90% confidence, one needs 10 ±3
years of monthly mean wind-speed records.

Moreover, raising the confidence level extends the minimum length of wind-speed data to compute
RCoV. At 95% confidence level, the median convergence years is 20 years, and 2.5% of grid points in
the CONUS require more than 37 years of monthly mean data to calculate RCoV (Fig. 9b and Table B5).
Additionally, using yearly mean wind speeds, instead of monthly data, to calculate RCoV leads to much
longer convergence time. At 95% confidence, 33 years of annual-mean data is the average required length,



and half of the CONUS grid points have convergence years over 37 years (Fig. 9b and Table B5). We

also perform the same analysis on CoV and $\sigma$ of wind speeds (Table B5). Although CoV and $\sigma$ result in

shorter convergence years, these non-robust and non-resistant methods yield worse correlations between

wind-speed and energy-production variabilities than RCoV, and hence we focus on demonstrating the

RCoV results.

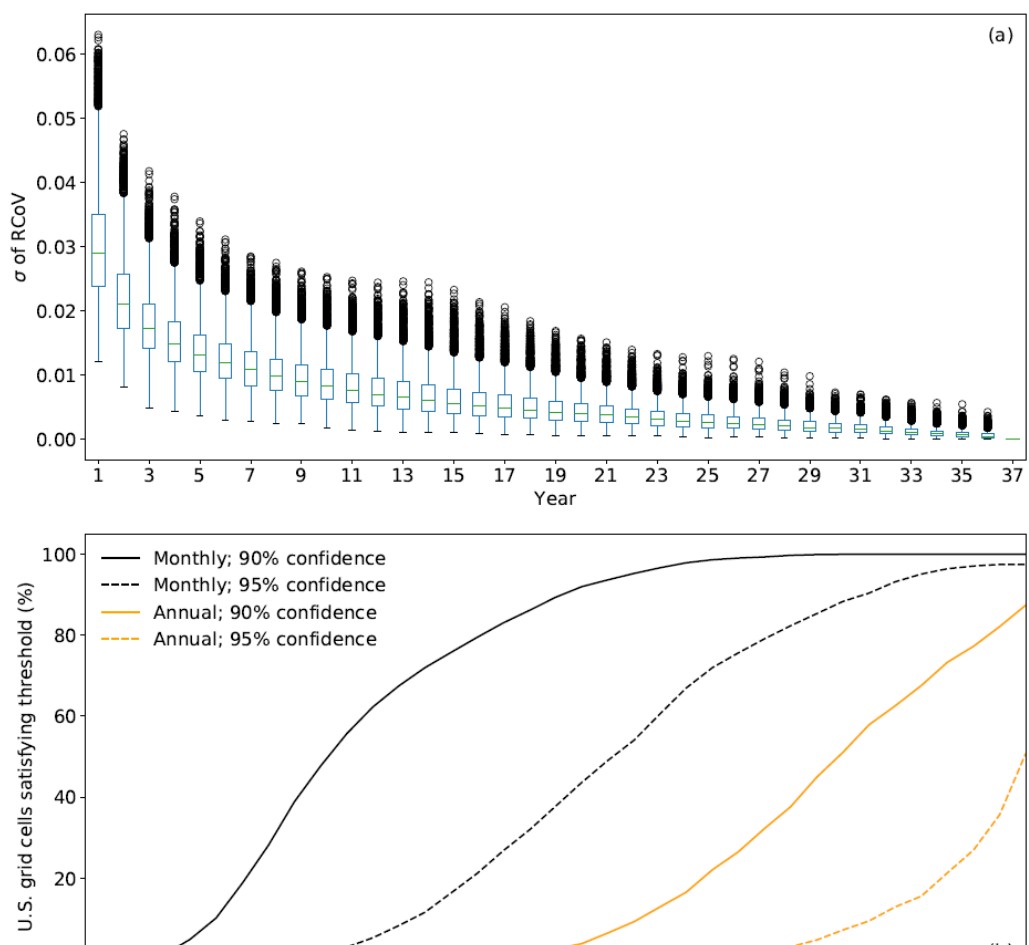




**Figure 9: (a) Boxplots of σ's of wind-speed RCoVs, where the RCoVs are calculated using monthly mean MERRA-2 data of 1 to 37 years. For each year, each box summarizes the σ from each MERRA-2 grid cells in the CONUS; (b) The fraction of grid cells in the CONUS that the pair of the $\chi^2$-derived σ's from each of those grid cells become smaller than the 37-year σ. The solid black, dash black, solid orange, and dash orange lines respectively indicate the minimum length of data: when the wind-speed RCoV using**

**monthly mean data yields 10% deviation at maximum from the 37-year value at 90% confidence level; when the wind-speed RCoV using monthly mean data yields 5% deviation at maximum from the 37-year value at 95% confidence level; when the wind-speed RCoV using yearly mean data yields 10% deviation at maximum from the 37-year value at 90% confidence level; and when the wind-speed RCoV using yearly mean data yields 5% deviation at maximum from the 37-year value at 95% confidence level.**

Spatial distributions of wind-speed RCoVs across the CONUS identify locations with reliable wind resources. Based on the site-specific convergence years at 90% confidence level (Fig. 10a), we calculate the RCoVs with monthly mean wind speeds of the particular time spans at each grid point and normalize with the CONUS median (Fig. 10b). Regions requiring long wind-speed records irregularly scatter across the continent, such as the Northeast, the Dakotas, and Texas. The mountainous states generally illustrate

high RCoVs, including the Appalachians and the Rockies. Given the strong correlations between the wind-speed RCoV and energy-production RCoV, Fig. 10b offers a realistic estimation of the general spatial pattern of the variability in wind-energy production as well. Note that qualitatively, Fig. 10b is similar to the maps of wind-speed variability in Figure 13a of Gunturu and Schlosser (2012) and in Figure 3 in Hamlington et al. (2015), which also illustrate the variability of wind resources in the CONUS. In

addition, using a fixed length of wind-speed data of 10 years for all CONUS grid points to compute RCoV results in a nearly identical spatial distribution to the pattern in Fig. 10b.

   Further, an ideal location for wind farms should exhibit ample wind speeds with low variability. We combine the spatial variations of the normalized RCoV and the long-term wind resource (Fig. 10b and c), and we differentiate regions according to the CONUS median RCoV and wind speed (Fig. 10d).

Favourable candidates for wind-farm developments have above-average wind speeds and below-average variabilities, such as the Plains, parts of the Upper Midwest, spots in the Columbia River region and the Carolinas; poor places for wind power with weak winds and strong variabilities include the Appalachians and most of the Northeast.

   The convergence years in some CONUS grid points are beyond 37 years when we increase the

confidence level from 90% to 95% (Fig. 9b and Table B5), and those grid points do not demonstrate any



geographical pattern as in Fig. 10a. Additionally, when using RCoV to represent IAV, the spatial patterns of required data lengths and the resultant normalized RCoVs for annual data are notably different from the monthly mean results, and geographical features seem to be irrelevant (Fig. A3). Furthermore, the categorical features of CoV resemble those of RCoV for onshore wind resources in the CONUS, whereas 575  using $\sigma$ results in notably distinct classifications of CONUS wind resources (Fig. 10d and Fig. A4).

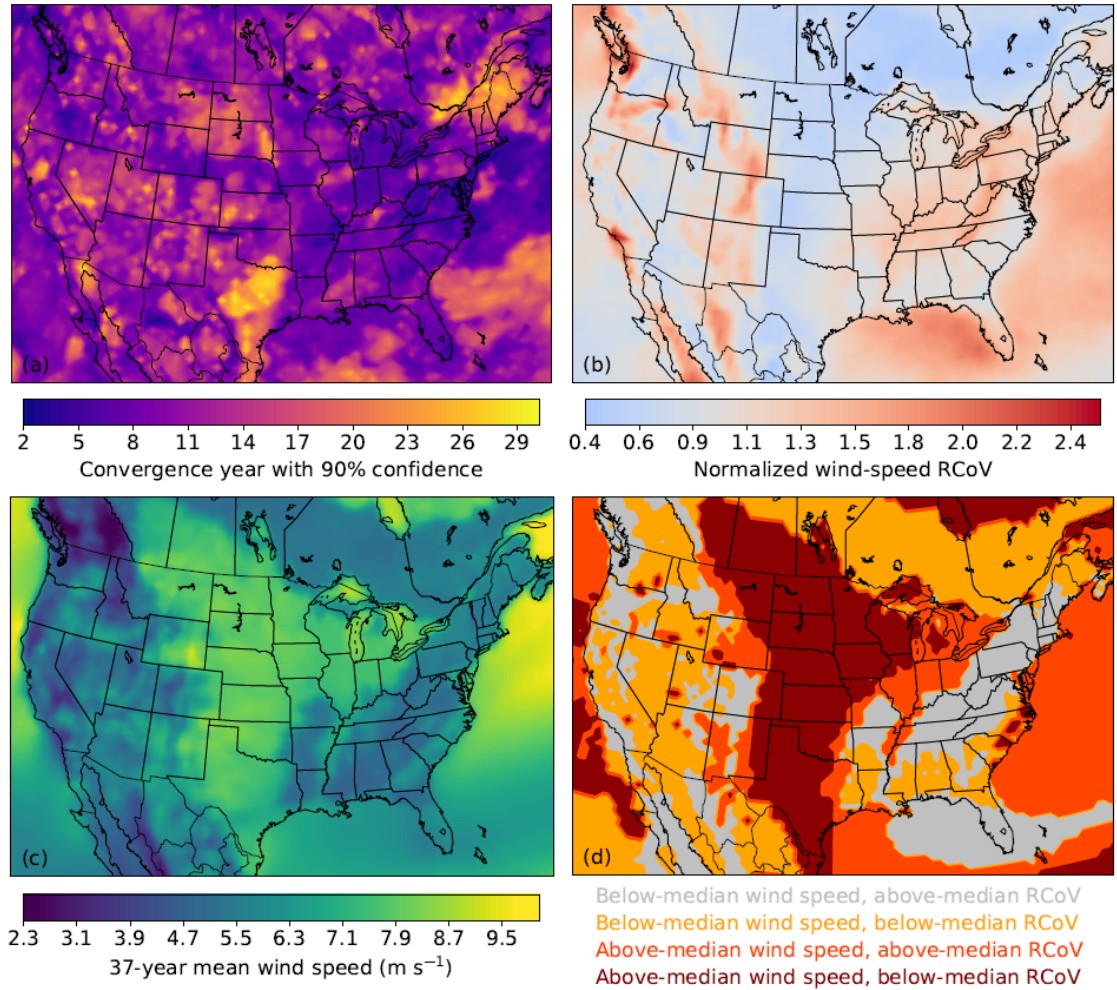



**Figure 10: (a) Map of the convergence years, or years of monthly mean wind-speed data required to derive a maximum of 10%**
**deviation from the 37-year RCoV at each grid point, at 90% confidence level. The CONUS median is 10 years with the MAD of 3**
**years; (b) Map of RCoV of monthly mean wind speed using the grid-cell-specific convergence years in (a), normalized using the**
**CONUS RCoV median at 0.100. The RCoVs illustrated are averaged over (37-convengence year+1) available year blocks. The MAD**
**of the normalized RCoV in the CONUS is 0.224; (c) Map of the mean monthly wind speed at 80 m of 37 years from 1980 to 2016.**
**The CONUS median is 6.45 m s⁻¹ with the MAD of 1.03 m s⁻¹; (d) Map of wind resource and its variability, by summarizing (b) and**
**(c) into four categories: regions with below-median wind speed and above-median RCoV (grey), regions with below-median wind**
**speed and below-median RCoV (orange), regions with above-median wind speed and above-median RCoV (orange red), and regions**
**with above-median wind speed and below-median RCoV (dark red), based on the CONUS median wind speed and RCoV.**

## 4 Discussion

When using statistically robust and resistant variability metrics, higher correlations between variabilities of wind speed and energy production emerge. Statistically robust methods do not assume or
require any underlying wind-speed distributions, and statistically resistant methods are insensitive to wind-speed extremes. Of all methods, three robust and resistant metrics, RCoV, MAD divided by trimean and IQR divided by median, result in the largest three $r$'s in Table 3 and Table B1, suggesting them as the most useful metrics to quantify long-term variability. Depending on the meteorological-data availability, wind-speed characteristics, and terrain complexity, different methods are appropriate in
different conditions. Nevertheless, robust and resistant methods are best able to relate wind-speed variability and energy-generation variability, and RCoV is the most effective one among all.

Overall, of all methods, RCoV consistently yields the strongest correlations between wind-speed and energy variabilities and exhibits reasonable asymptote periods (Table 3 and Table B1), even after accounting for random standard errors and modifying the $R^2$ and $r$ thresholds (Table B3). In addition,
assessing wind-speed RCoV with 90% confidence requires 10 ±3 years of wind-speed data (Fig. 9 and Table B5), which exceeds the asymptote periods of 2 to 6 years to yield strong wind-speed and energy-production correlations (Table 3). Even though different locations require various spans of data (Fig. 10a), the average of the resultant RCoVs using 10 years of wind speeds leads to nearly identical spatial distributions (Fig. 10b). Therefore, to effectively quantify wind-speed variability and thus to adequately
derive energy-generation variability, we recommend using the RCoV with 10 years of monthly mean wind-speed data.



Annual-mean data are inadequate to relate wind-speed and energy-production IAVs or to represent wind-speed IAVs. We cannot determine the minimum years of data to relate annual wind-speed and energy IAVs because their correlations decline with the length of data (Fig. 7). Moreover, the coarse time resolution of annual averages smooths out fluctuations of smaller time scales. Yearly mean wind speeds also possess different distribution characteristics, such as skewness and kurtosis, compared to those of finer temporal resolutions (Lee et al., 2018). The non-zero kurtosis and skewness in Table 2 and in Lee et al. (2018) illustrate that most of the distributions of annual-mean wind speeds in the CONUS are non-Gaussian. Hence, using non-robust metrics, such as $\sigma$, to evaluate IAV with samples of annual means from non-Gaussian distributions can lead to incorrect representation of variability.

Additionally, extended years of wind-speed data are also necessary to compute RCoV and represent IAV (Fig. A3a), and the resultant IAVs (Fig. A3b) differ from the variabilities calculated via monthly wind speeds (Fig. 10b). For instance, the low IAVs in the Appalachians (Fig. A3b) calculated with yearly mean wind speeds contradict the pattern of high monthly mean wind-speed RCoVs in mountainous areas (Fig. 10b) as well as the findings in past research (Gunturu and Schlosser, 2012; Hamlington et al., 2015). Furthermore, some of the grid points require more than 37 years of yearly mean data to calculate wind-speed RCoV with statistical confidence (Fig. 9 and Table B5). Although RCoV does not yield the strongest 37-year $r$ in relating wind-speed and energy IAVs, readers should be cautious when using a limited number of annual-mean data to derive IAVs. In short, to effectively assess the long-term variability of wind-farm productivity, one should use wind speeds finer than yearly mean data.

Regions with ample wind resources and low variability favour wind-energy developments, coinciding the locations of many existing wind farms in the CONUS (Fig. 10d). Wind farms in the Plains and parts of the upper Midwest benefit from the above-average wind speeds and the below-average wind-speed RCoVs. Other regions, such as segments in the Columbia River region and the Carolinas, also experience strong, consistent winds. The Northeast and the Appalachians are relatively unfavourable for producing stable onshore wind-energy supply, whereas the area east of Cape Cod in Massachusetts and the sections along the West Coast exhibit promising offshore wind resource. Wind-farm developers should account for wind resource as well as its long-term variability in repowering existing turbines and building new wind farms.





Furthermore, mathematically, a normalized spread metric, namely a spread statistic divided by an average metric, is more useful than solely a spread metric in assessing variability, and a normalized spread metric should always be presented with the corresponding averaging metric. For example, RCoV and CoV between wind speed and energy yield larger $r$'s than MAD and $\sigma$ (Table 3 and Fig. A1), and the $r$'s between wind-speed RCoV and CoV are also higher than those comparisons involving MAD and $\sigma$ (Fig.

6). For $\sigma$, the root-mean-square of the deviation from the mean, is not statistically robust or resistant, and 1 $\sigma$ means the uncertainty is 18.3% from the mean. Hence, CoV, or the $\sigma$ divided by the mean, is the respective normalized uncertainty metric to $\sigma$. For instance, the wind-speed CoVs of both the OR and TX sites are about 0.13 (Table 2), implying the $\sigma$ is 13% from the mean. In contrast, using RCoV, or the MAD divided by the median, is a robust and outlier-resistant metric of normalized uncertainty. For

example, the wind-speed RCoV of the OR and TX sites are 0.08 and 0.09 respectively (Table 2), indicating the MADs are 8% and 9% from their median wind speeds. Even though RCoV is not as commonly used and not as intuitive as $\sigma$ or CoV, RCoV is unrestricted by any underlying-distribution assumptions. Overall, to correctly and effectively use the normalized spread metrics, both the normalized spread metric and the average value need to be stated clearly in pairs. In other words, the interpretation

of "variability is 2%" oversimplifies the statistics of uncertainty quantification. Therefore, we recommend presenting both the RCoV and the median of a time series together in estimating variability.

       Distribution diagnostics, other than the variability metrics, are also effective in identifying the characteristics of wind-energy production. We examine distribution parameters resulting in strong wind-speed-energy correlations, including kurtosis and YKI (Table 3 and Table B2), which assess the degree

of deviations from a Gaussian distribution. For instance, we confirm the monthly and annual wind-speed distributions for our case studies in OR and TX are not perfectly Gaussian because of their non-zero kurtosis and skewness values (Table 2), as well as their portions of data within 1 σ. Moreover, a multi-modal or an asymmetric wind-speed distribution (Fig. 3c and d) also implies a non-Gaussian energy production distribution. Gaussian distribution is invalid for wind speeds across averaging time scales in

general (Lee et al., 2018). Hence, understanding the underlying distribution of wind resources can validate the applications and the legitimacy of Gaussian statistics, especially in quantifying P50 and the associated losses and uncertainties.





**5 Conclusions**

Wind-speed variability is a crucial component in assessing the overall uncertainty of P50, and this
study highlights the importance of using rigorous methods to estimate inter-monthly and inter-annual
variability. To search for suitable ways to quantify this uncertainty under different conditions, we
investigate 27 combinations of spread metrics over 607 wind farms in the US, with closer examination of
two geographically-distinct sites. We evaluate the methods for robustness to non-Gaussian distributions
and resistance to extreme values, in contrast to the common practice of using only standard deviation ($\sigma$).
We calculate variabilities using monthly and annual mean wind speeds from the Modern-Era
Retrospective Analysis for Research and Applications, Version 2 (MERRA-2) reanalysis dataset and
wind-farm monthly net energy productions from the Energy Information Administration (EIA). We find
that within the contiguous United States (CONUS), statistically robust and resistant methods predict
variabilities more accurately, particularly in that wind-speed variabilities strongly correlate with observed
energy-production variabilities.

We recommend Robust Coefficient of Variation (RCoV) to quantify variabilities of wind resource
and energy production. RCoV, defined as the median of absolute deviation from the median wind speed
divided by the median of the wind speed, is a robust and resistant spread metric, in contrast to $\sigma$. RCoV
yields strong correlations consistently (a Pearson's $r$ of 0.856 with 37 years of monthly means) in various
sensitivity tests via different correlation coefficients, whereas $\sigma$ does not. In other words, using RCoV, a
wind farm with high wind-speed fluctuations also possesses high variations in wind-energy generations
and vice versa, whereas other metrics do not reflect that relationship as effectively. RCoV, as a normalized
spread metric, also leads to a more accurate depiction of wind-speed variabilities than $\sigma$, a simple spread
metric. Contrary to the custom of displaying uncertainty in one percentage value, we advise users to assess
both the RCoV and the median in estimating inter-monthly variability. Moreover, depending on the
location, on average 10 ±3 years of monthly wind-speed data is necessary to compute wind-speed RCoV
with 90% statistical confidence, such that the resultant RCoV deviates within 10% of the long-term
RCoV.

RCoV characterizes the spreads of the distributions of wind resources and wind-energy productions.
The relatively low monthly mean wind-speed RCoVs in the central U.S. indicate stable long-term wind




resources, and the RCoV overall spatial distribution in the CONUS agrees with the findings from past research. Other distribution diagnostics, such as kurtosis and skewness, also result in high correlations between monthly mean wind speed and energy generation, and thus they adequately represent energy-production characteristics.

Because the long-term correlations between the wind-speed and energy-production inter-annual variabilities (IAVs) are weak (a Pearson's $r$ of 0.668 for RCoV with 37 years of data) and decrease with the length of data, we also do not recommend readers to calculate variabilities with annual-mean data. Hence, we cannot determine the minimum length of annual-mean data required for skilful assessment of IAV. Although the concept of IAV has been essential in determining the annual energy production in the

wind resource assessment process, annual-mean wind speeds mask signals of finer temporal scales and thus lead to unreliable representations of long-term variability. Overall, uncertainty arises in the process of calculating IAVs based on limited samples, whereas RCoV yields credible inter-monthly variabilities considering the adequate amount of monthly mean data.

Now that we have highlighted the preferred structure of using RCoV, we can assess finer-scale

variations using high-resolution wind-speed and energy-production data. With data of different temporal scales, the autocorrelation of wind resources and its relationship with long-term energy-production variations can also be quantified. The influence of climatic cycles on energy production can be explored. Furthermore, applying the concept of RCoV to reduce the uncertainty of P50 and assist financial decisions can be beneficial to the industry.

**Data Availability**

The MERRA-2 data and the EIA data used in this study are publicly available at disc.sci.gsfc.nasa.gov/ and www.eia.gov/renewable.



**Appendix A**

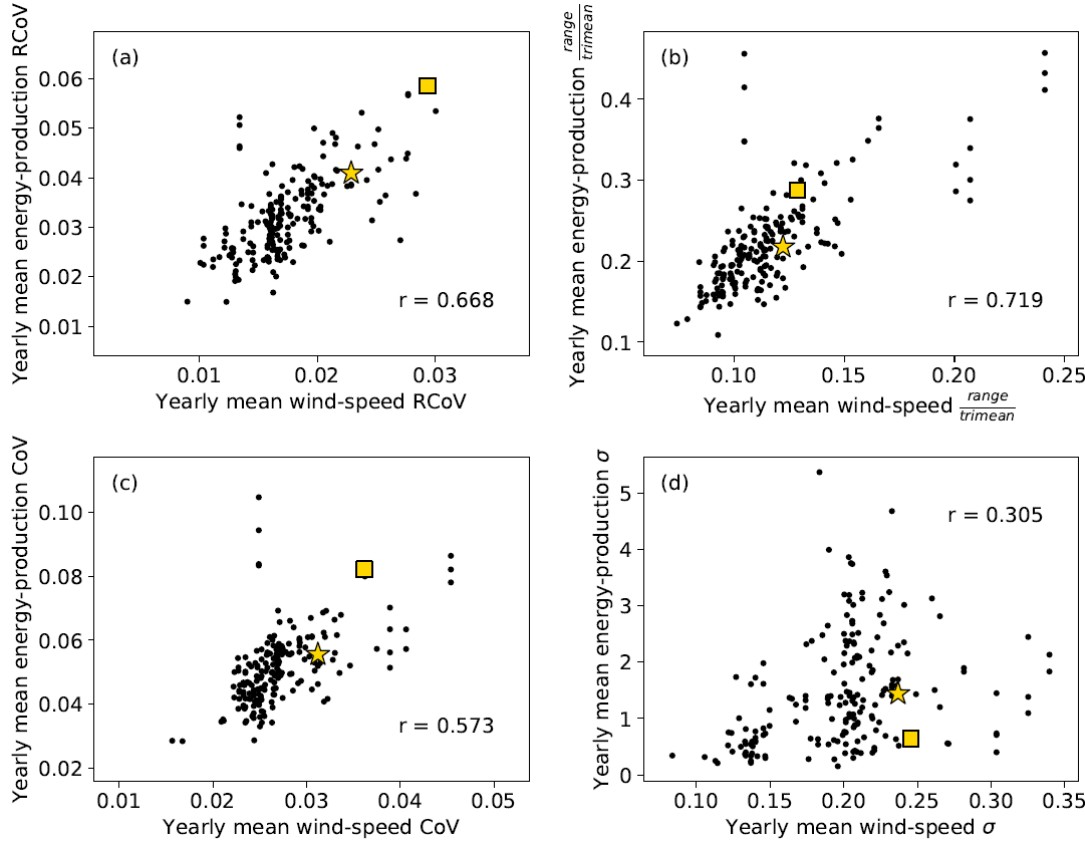

Figure A 1: As in Fig. 4, but the metrics are calculated using annual-mean wind speed and energy production.





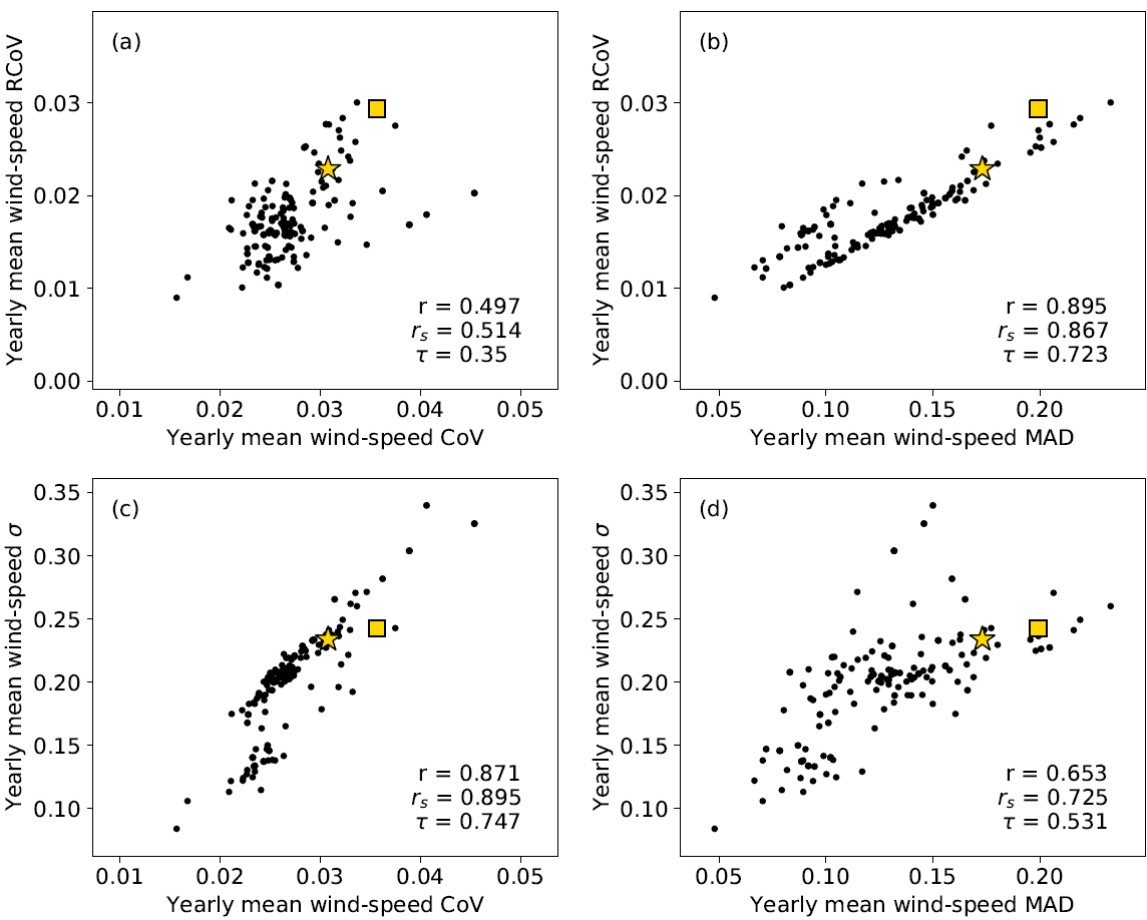


**Figure A 2: As in Fig. 6, but the metrics are calculated using yearly mean wind speed.**





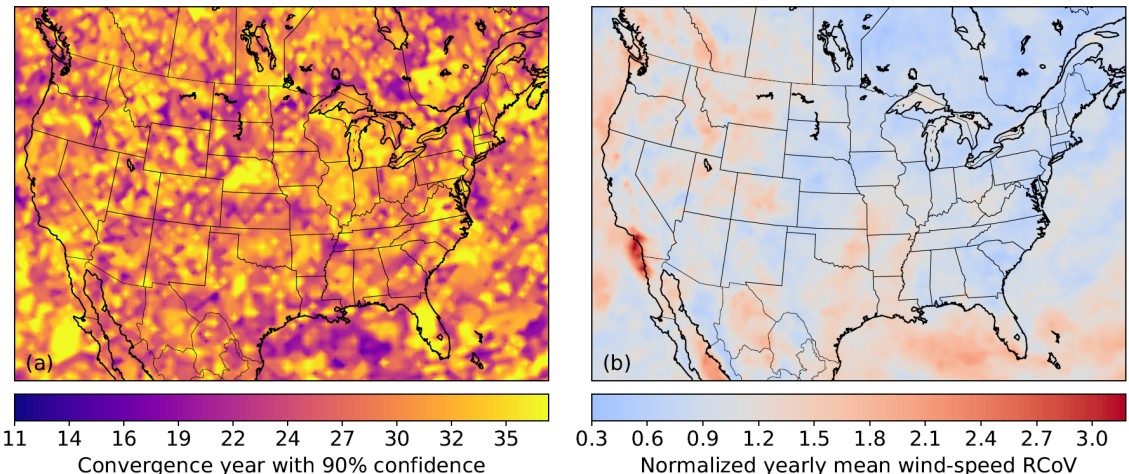

**Figure A 3: As in Fig. 10a and b, but the data plotted are annual-mean wind speeds: (a) Map of the convergence years, or years of**
**wind-speed data required to derive a maximum of 10% deviation from the 37-year RCoV at each grid point, at 90% confidence**
**level. Because 12.6% of the CONUS grid points yield convergence years beyond 37 years using annual data (solid orange line in Fig.**
**9 and first column in Table B5), we assign 37 years as the convergence years for those grid points. After excluding the non-numeric**
**values, the CONUS median is 27 years and the MAD is 4 years; (b) Map of RCoV of annual-mean wind speed using the grid-cell-**
**specific convergence years in (a), normalized using the CONUS RCoV median at 0.020. The RCoVs illustrated are averaged over**
**(37-convergence year+1) available year blocks. The MAD of the normalized RCoV in the CONUS is 0.205.**





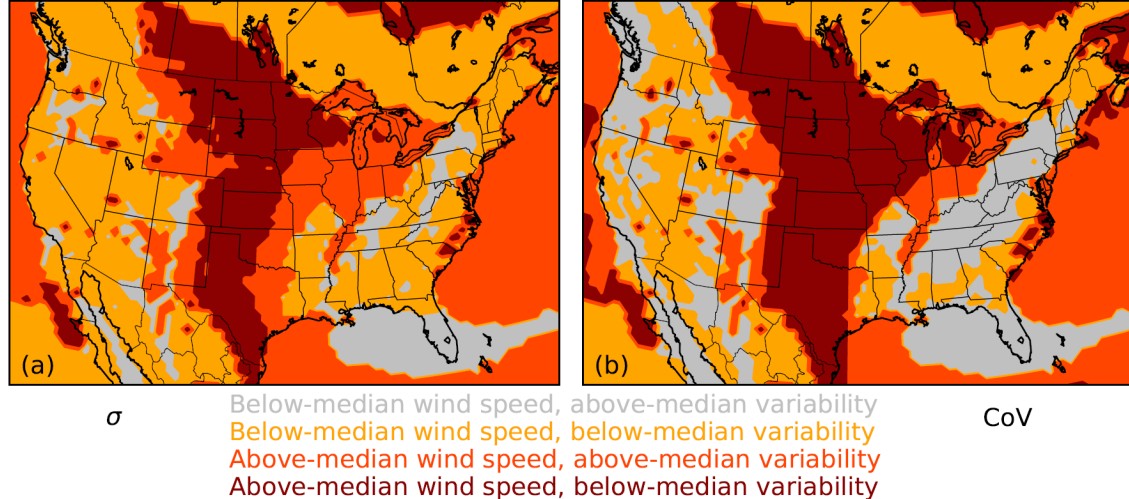

Below-median wind speed, above-median variability
Below-median wind speed, below-median variability
Above-median wind speed, above-median variability
Above-median wind speed, below-median variability

**Figure A 4:** As in Fig. 10d, but the spread metrics are (a) σ and (b) CoV, calculated using monthly mean wind speeds of 37 years.




## Appendix B

**Table B 1: Description of the 26 spread metrics tested, adapted from (Wilks, 2011), and the 37-year $r$'s from the r-filtered monthly data. $q_{0.25}$ is the 25th percentile (first quartile), $q_{0.5}$ is the 50th percentile (median), and $q_{0.75}$ is the 75th percentile (third quartile). $Trimean = \frac{1}{4}(q_{0.25} + 2 \times q_{0.5} + q_{0.75})$, $range(x) = \max(x) - \min(x)$, and an overbar ($\bar{x}$) indicates the arithmetic mean. Reason I: the metric is not robust because the metric possesses distribution constraints which is usually Gaussian, and the metric is not resistant because outliers influence it; Reason II: the metric is not resistant as outliers influence it; Reason III: the numerator of the metric is not robust or resistant; Reason IV: the denominator of the metric is not robust or resistant; Reason V: the numerator of the metric is not resistant.**

| Spread metrics | 37-year $r$ | Robust and resistant | Why not robust and resistant |
|---|---|---|---|
| Interquartile range $(IQR) = q_{0.75} - q_{0.25}$ | 0.214 | Yes | / |
| $\dfrac{IQR}{median}$ | 0.845 | Yes | / |
| $\dfrac{IQR}{trimean}$ | 0.834 | Yes | / |
| Median deviation from median $= median[x - median(x)]$ | -0.048 | Yes | / |
| Median Absolute Deviation $(MAD)$ $= median|x - median(x)|$ | 0.196 | Yes | / |
| Robust Coefficient of Variation $(RCoV) = \dfrac{MAD}{median}$ | 0.856 | Yes | / |
| Exponential $RCoV = \dfrac{\ln(MAD)}{\ln(median)}$ | 0.595 | Yes | / |
| $\dfrac{MAD}{trimean}$ | 0.848 | Yes | / |
| Standard deviation $(\sigma) = \sqrt{\dfrac{1}{n-1}\sum_{i=1}^{n}(x_i - \bar{x})^2}$ | 0.184 | No | Reason I |
| Variance $(\sigma^2) = \dfrac{1}{n-1}\sum_{i=1}^{n}(x_i - \bar{x})^2$ | 0.136 | No | Reason I |




| | | | |
|---|---|---|---|
| $Coefficient\ of\ Variation\ (CoV) = \dfrac{\sigma}{mean}$ | 0.704 | No | Reason I |
| $Exponential\ CoV = \dfrac{\ln{(\sigma)}}{\ln{(mean)}}$ | 0.466 | No | Reason I |
| $Mean\ deviation\ from\ mean = \overline{(x-\bar{x})}$ | -0.043 | No | Reason I |
| $Mean\ Absolute\ Deviation = \overline{|x-\bar{x}|}$ | 0.187 | No | Reason I |
| $Trimmed\ standard\ deviation\ (\sigma)$ $= standard\ deviation\ without\ values\ below\ Q10\ and\ Q90,$ $= \sqrt{\dfrac{1}{n-2k}\displaystyle\sum_{i=k+1}^{n-k}\left(x_{(i)}-\bar{x}_a\right)^2}, k\ as\ the\ nearest\ integer\ to\ a$ $\times n$ | 0.206 | No | Reason I |
| $\dfrac{Trimmed\ \sigma}{\bar{x}}$ | 0.775 | No | Reason I |
| $Range$ | 0.177 | No | Reason II |
| $\dfrac{Range}{\bar{x}}$ | 0.609 | No | Reason I |
| $Seasonality\ Index = \dfrac{\sum|x-\bar{x}|}{n \times \bar{x}}$ (modified from Walsh and Lawler (1981)) | 0.744 | No | Reason I |
| $\dfrac{\sigma}{median}$ | 0.743 | Partially | Reason III |
| $\dfrac{\sigma}{trimean}$ | 0.728 | Partially | Reason III |
| $\dfrac{IQR}{\bar{x}}$ | 0.818 | Partially | Reason IV |
| $\dfrac{MAD}{\bar{x}}$ | 0.834 | Partially | Reason IV |
| $\dfrac{Trimmed\ \sigma}{median}$ | 0.806 | Partially | Reason III |
| $\dfrac{Trimmed\ \sigma}{trimean}$ | 0.794 | Partially | Reason III |



| | | | |
|---|---|---|---|
| $\dfrac{Range}{median}$ | 0.650 | Partially | Reason V |
| $\dfrac{Range}{trimean}$ | 0.635 | Partially | Reason V |



**Table B 2: Description of the distribution diagnostics tested, adapted from (Wilks, 2011) and the 37-year $r$'s from the r-filtered monthly data. Reason I: the metric is not robust because the metric possesses distribution constraints which is usually Gaussian, and the metric is not resistant because outliers influence it; Reason II: the metric is not robust because it assumes Weibull distribution.**

| Other diagnostics | Description | 37-year $r$ | Robust and resistant | Why not robust and resistant |
|---|---|---|---|---|
| $Kurtosis\ (Tailedness)$ $= \dfrac{\frac{1}{n}\sum_{i=1}^{n}(x_i - \bar{x})^4}{(\frac{1}{n}\sum_{i=1}^{n}(x_i - \bar{x})^2)^2}$ | Positive value means the distribution is tail-heavy with more and more extreme outliers compared to Gaussian; vice versa | 0.936 | No | Reason I |
| $Skewness$ $= \dfrac{\frac{1}{n}\sum_{i=1}^{n}(x_i - \bar{x})^3}{(\frac{1}{n}\sum_{i=1}^{n}(x_i - \bar{x})^2)^{\frac{3}{2}}}$ | Positive value means long right tails, or right-skewed; vice versa | 0.943 | No | Reason I |
| $Yule - Kendall\ Index\ (YKI)$ $= \dfrac{q_{0.25} - 2 \times q_{0.5} + q_{0.75}}{IQR}$ | Positive value means long right tails, or right-skewed; vice versa | 0.778 | Yes | / |
| $Weibull\ scale\ parameter$ | Determine the peak and the stretch | 0.379 | No | Reason II |
| $Weibull\ shape\ parameter$ | Determine the average, the symmetry and the shape | 0.721 | No | Reason II |
| $Autocorrelation$ | Pearson's $r$ with its own past and future values | Not applicable | Not applicable | / |





**Table B 3: As in Table 3, but calculated metrics, the associated correlations and asymptote periods using different R² and _r_ filters and adding random standard error to predicted monthly total energy productions. The sample sizes of the 0.7-_r_ threshold test, the 0.9-_r_ threshold test and the random error tests are 306, 83, and 195 wind farms respectively.**

| Sensitivity test | $R^2 = 0.6$ $r = 0.7$ | | $R^2 = 0.85$ $r = 0.9$ | | Random error | |
|---|---|---|---|---|---|---|
| Spread metrics | 37-year $r$ | Asymptote years | 37-year $r$ | Asymptote years | 37-year $r$ | Asymptote years |
| CoV | 0.650 | 6 | 0.787 | 3 | 0.675 | 6 |
| $\frac{\sigma}{median}$ | 0.682 | 5 | 0.820 | 2 | 0.708 | 4 |
| $\frac{\sigma}{trimean}$ | 0.671 | 5 | 0.804 | 3 | 0.695 | 5 |
| $\frac{IQR}{mean}$ | 0.786 | 4 | 0.837 | 3 | 0.774 | 7 |
| $\frac{IQR}{median}$ | 0.811 | 3 | 0.865 | 2 | 0.799 | 6 |
| $\frac{IQR}{trimean}$ | 0.801 | 4 | 0.851 | 3 | 0.789 | 7 |
| RCoV | 0.815 | 3 | 0.879 | 2 | 0.808 | 6 |
| $\frac{MAD}{mean}$ | 0.793 | 3 | 0.859 | 3 | 0.786 | 7 |
| $\frac{MAD}{trimean}$ | 0.807 | 3 | 0.870 | 3 | 0.800 | 6 |
| $\frac{Range}{mean}$ | 0.524 | 31 | 0.767 | 26 | 0.567 | 29 |
| $\frac{Trimmed\ \sigma}{median}$ | 0.736 | 5 | 0.816 | 3 | 0.741 | 6 |
| $\frac{Trimmed\ \sigma}{trimean}$ | 0.753 | 4 | 0.831 | 3 | 0.758 | 5 |
| Seasonality Index, modified from Walsh and Lawler (1981) | 0.695 | 5 | 0.804 | 3 | 0.710 | 5 |
| Other diagnostics | | | | | | |
| Kurtosis | 0.896 | 5 | 0.927 | 1 | 0.886 | 14 |
| Skewness | 0.931 | 1 | 0.951 | 1 | 0.918 | 8 |
| YKI | 0.756 | 23 | 0.833 | 19 | 0.669 | 25 |
| Weibull shape parameter | 0.656 | 5 | 0.802 | 3 | 0.706 | 4 |





**Table B 4: As in Table 3, but calculated metrics, the associated correlations and asymptote periods using annual-mean wind speed and energy production using the 195 r-filtered sites.**

| IAV metrics | 37-year $r$ | Asymptote years |
|---|---|---|
| CoV | 0.573 | 27 |
| $\dfrac{\sigma}{median}$ | 0.567 | 27 |
| $\dfrac{\sigma}{trimean}$ | 0.569 | 27 |
| $\dfrac{IQR}{mean}$ | 0.699 | 24 |
| $\dfrac{IQR}{median}$ | 0.697 | 24 |
| $\dfrac{IQR}{trimean}$ | 0.699 | 24 |
| RCoV | 0.668 | 27 |
| $\dfrac{MAD}{mean}$ | 0.670 | 25 |
| $\dfrac{MAD}{trimean}$ | 0.670 | 25 |
| $\dfrac{Range}{mean}$ | 0.723 | 27 |
| $\dfrac{Trimmed\ \sigma}{median}$ | 0.567 | 27 |
| $\dfrac{Trimmed\ \sigma}{trimean}$ | 0.569 | 27 |
| Seasonality Index, modified from Walsh and Lawler (1981) | 0.547 | 29 |
| Other diagnostics | | |
| Kurtosis | 0.985 | 5 |
| Skewness | 0.980 | 4 |
| YKI | 0.853 | 12 |
| Weibull shape parameter | 0.649 | 28 |





**Table B 5: Convergence years based on the $\chi^2$ approach of wind-speed RCoV (as in Fig. 8 and 9), wind-speed CoV, and wind-speed σ, using monthly and yearly wind speeds. The calculations of median and MAD exclude the data with convergence years beyond 37 years in the CONUS.**

| Monthly mean wind speed | RCoV | | CoV | | σ | |
|---|---|---|---|---|---|---|
| Confidence level | 90% | 95% | 90% | 95% | 90% | 95% |
| 37-year sample size (of 5049 grid points) | 5049 | 4923 | 5049 | 5039 | 5049 | 5048 |
| Convergence years – CONUS median | 10 | 20 | 4 | 12 | 4 | 12 |
| Convergence years – CONUS MAD | 3 | 4 | 2 | 5 | 2 | 5 |
| Convergence years – OR site | 12 | 20 | 6 | 15 | 6 | 15 |
| Convergence years – TX site | 25 | 31 | 7 | 24 | 5 | 24 |
| Yearly mean wind speed | RCoV | | CoV | | σ | |
| Confidence level | 90% | 95% | 90% | 95% | 90% | 95% |
| 37-year sample size (of 5049 grid points) | 4414 | 2565 | 5034 | 4292 | 5034 | 4301 |
| Convergence years – CONUS median | 27 | 33 | 20 | 28 | 19 | 28 |
| Convergence years – CONUS MAD | 4 | 2 | 4.5 | 3 | 4 | 3 |

**Acknowledgements**

The Alliance for Sustainable Energy, LLC (Alliance) is the manager and operator of the National Renewable Energy Laboratory (NREL). NREL is a national laboratory of the U.S. Department of Energy, Office of Energy Efficiency and Renewable Energy. This work was authored by the Alliance and supported by the U.S. Department of Energy under Contract No. DE-AC36-08GO28308. Funding was provided by the U.S. Department of Energy Wind Energy Technologies Office.

The U.S. Government retains and the publisher, by accepting the article for publication, acknowledges that the U.S. Government retains a nonexclusive, paid up, irrevocable, worldwide license to publish or reproduce the published form of this work, or allow others to do so, for U.S. Government purposes.

    The authors would like to thank our collaborators, Vineel Yettella and Mark Handschy of the Cooperative Institute for Research in Environmental Sciences (CIRES) at the University of Colorado
Boulder, our colleagues at NREL especially Paul Veers, and Cory Jog at EDF Renewable Energy.



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
