# Peer review of "Assessing Variability of Wind Speed: Comparison and Validation of 27 Methodologies"

_Wind Energy Science, 2018_

## Referee Comment (RC1) · Anonymous Referee #1 · 19 Aug 2018

Overall I find this work to be a substantial project and a valuable contribution to wind energy and wind climatology communities. I have only a few comments that I ask the authors to consider regarding this manuscript and perhaps their future work.

(1) Because the IAV statistical results are based on your filtered data (e.g., Section 2.2), more explanation/justification of your methods would be useful. For example, why use linear regression when power is a nonlinear function of wind speed? What exactly are your criteria for identifying "underproduction for reasons other than low wind speed" and "potentially erroneous overproduction" (lines 132-134), and how confident are you that these are legitimate outliers? What is the proportion of "derived energy data" included in each of the time series for the 204 stations that require such data (line 143)?

(2) Fig 1: Given the geographic distribution of retained sites, is there a need to consider geographically weighting the analysis results so that the central Plains results (for example) are not unduly influencing your interpretation of the statistics?

(3) Fig 2b, c: What would these figures look like if plotted with the R2- and r-filtered data?

(4) Fig 6b, c: What are the characteristics of those sites that parallel the "line" that goes through the TX site? What makes them not deviate so much on panels a, d? Are these the same sites that show this pattern in Fig A2 b, c?

(5) I am a fan of MAD-based statistics but not necessarily to the exclusion of other types of statistics. It would be helpful and interesting to include some discussion on why the different metrics give different results and how they may highlight different aspects of what the wind speeds are like at these stations (for example, in reference to the Oregon site in line 382). You do acknowledge the potential utility of different measures in the Discussion, lines 593-596, but the paper itself seems to be focused on identifying "the one" measure that should be used. Is that your explicit intention?

---

## Referee Comment (RC2) · Anonymous Referee #2 · 28 Aug 2018

Excellent work and paper. My minor comment is with respect to the use of reanalysis data which may have a lower interannual variability than actual site data. A short comment by the authors in the paper could address this quite easily.

---

## Author Comment (AC1) · 4 Oct 2018

Because the font and font size is fixed (in plain text) as set by WES, for clarity, the reviewer comments are numbered, and the first paragraphs of our responses open with "——".

REVIEWER 1 COMMENTS:

(1) Overall I find this work to be a substantial project and a valuable contribution to wind energy and wind climatology communities. I have only a few comments that I ask the authors to consider regarding this manuscript and perhaps their future work.

—— We thank the reviewer for the comments and suggestions.

[Figure]

(2) Because the IAV statistical results are based on your filtered data (e.g., Section 2.2), more explanation/justification of your methods would be useful. For example, why use linear regression when power is a nonlinear function of wind speed? What exactly are your criteria for identifying "underproduction for reasons other than low wind speed" and "potentially erroneous overproduction" (lines 132-134), and how confident are you that these are legitimate outliers? What is the proportion of "derived energy data" included in each of the time series for the 204 stations that require such data (line 143)?

—— We also perform a third-order polynomial fit to test the nonlinear relationship between wind speed and power production, and we find very similar results to the linear filtering, so we choose to focus on the linear regressions in the manuscript. The description of the polynomial test is now included in lines 137 to 139:

"We also apply a third-order polynomial fit (Archer and Jacobson, 2013), and it leads to very similar results to the linear model. Hence, we focus on presenting the results from the linear fit in this study."

The results from the polynomial and the linear fits are similar partly because wind speed is the only independent variable that is important (as mentioned in lines 193 to 195, air density is a trivial predictor.). Moreover, the data we use are monthly averaged wind speeds and monthly total energy production, so the third-order effect of wind speed on wind power (such as gusts) is also averaged out because of the coarse resolution of data.

The objective of the linear regression filtering process is to eliminate all the factors affecting power production other than wind speed. This process is also commonly used in the wind energy industry. To explain this explicitly, lines 142 to 144 now read:

"Through this filter, we ensure that wind speed is the primary driver of energy production in the wind farms with high $R^2$ values. Lunacek et al. (2018) also use a similar $R^2$-filtering method with a threshold of 0.7."

Assuming a Gaussian distribution of the energy production data at each site, using the 90% prediction interval would exclude the energy production below 1.64 times of the standard error (defined as underproduction) of the site-specific linear regression. Similarly, using the 99% prediction interval would exclude overproduction that are 2.58 times above the standard error. To quantify the confidence as well as the uncertainty associated with this method, we include the following in lines 135 to 137:

"In other words, we define the outliers of energy production using the threshold of 1.64 times below the standard error and 2.58 times above the standard error of the site-specific regression."

The attached RC1_Fig1.png (Fig. 1 here) is a histogram of derived energy data among the 349 R2-filtered sites. The median is 7.5 years.

To clearly describe the amount of energy data that are derived using linear fit, lines 148 to 150 now read:

"Of the 349 wind farms, 7.5 years is the median of the energy data that are derived via the linear fit, given the available EIA records between 2003 and 2016."

(3) Fig 1: Given the geographic distribution of retained sites, is there a need to consider geographically weighting the analysis results so that the central Plains results (for example) are not unduly influencing your interpretation of the statistics?

—— The goal of this study is to determine a holistic approach to evaluate wind-speed variability that is not geographically specific. Although many of the r-filtered sites locate in the Plains (Fig. 1), a nontrivial portion of the sites are scattered across the United States, therefore the r-filtered data are well represented geographically. The r-filtered points in Fig. 1 also represent the broad spatial distribution of wind sites with satisfying data quality.

Per the reviewer's comment, we do think exploring the geographical analysis of wind speed, wind-speed variability, and the relationship between wind speed and energy

production is an interesting future research topic. With improving quality and quantity of energy production data as well as the increasing number of new wind farms, we think the research is feasible in the near future.

(4) Fig 2b, c: What would these figures look like if plotted with the R2- and r-filtered data?

——– Please see the Fig2_S2.pdf attached (Fig. 2 here), and Fig. 2 in the paper is now updated. The R2-filtered and r-filtered data are the points above R2 (y-axes) of 0.75 in Fig. 2b and c.

(5) Fig 6b, c: What are the characteristics of those sites that parallel the "line" that goes through the TX site? What makes them not deviate so much on panels a, d? Are these the same sites that show this pattern in Fig A2 b, c?

——– The purpose of Fig. 6 (and Fig. A2) is to contrast the results of normalized spread metrics (particularly CoV and RCoV) and nonnormalized (or simple) spread metrics (particularly standard deviation and MAD). The data points that deviate from the line-like linear relationship between a normalized metric and a nonnormalized metric in Fig. 6b and c represent that the mean wind speeds of those sites are lower than the rest of the sites, when those sites possess the same magnitude of standard deviation or MAD. Hence, given the same standard deviation or MAD, the CoV or RCoV of each of those sites is lower than the others.

The data in Fig. 6a and d resemble a straight line, because they are contrasting a pair of normalized spread metrics and a pair of nonnormalized (or simple) spread metrics, respectively. The line-like feature in Fig. 6a and d is exactly what we expected, because the results from either normalized or nonnormalized metrics should be consistent. Similarly, the not-straight-line feature in Fig. 6b and c conveys that using normalized spread metrics can lead to different results than using nonnormalized spread metrics. This idea is discussed in lines 457 to 472.

We also confirm that those points in Fig. 6b and c located "out of the line" are also the same points in Fig. A2b and c.

(6) I am a fan of MAD-based statistics but not necessarily to the exclusion of other types of statistics. It would be helpful and interesting to include some discussion on why the different metrics give different results and how they may highlight different aspects of what the wind speeds are like at these stations (for example, in reference to the Oregon site in line 382). You do acknowledge the potential utility of different measures in the Discussion, lines 593-596, but the paper itself seems to be focused on identifying "the one" measure that should be used. Is that your explicit intention?

—— To quantify wind-speed variability without knowing the underlying distribution, we do recommend RCoV in general. Of course, different distribution metrics such as skewness, kurtosis, and lag-k correlations would also provide more information about the distribution itself. With such information, the analyst can then choose another appropriate spread metric, or even a collection of spread metrics, to assess the variability of wind speed of a location. The primary goal of this manuscript is to determine the most effective spread metric that is applicable for any locations with any distribution shapes, and thus we perform different analyses to support our suggestion on RCoV, such as correlating with energy production, the asymptote analysis, the chi-square test, etc. Throughout the manuscript, we also compare the results from nonrobust and nonresistant metrics, as well as nonnormalized metrics. Hence, in order to keep a sharp focus, we choose to exclude any in-depth discussion on how different metrics vary at particular locations.

In fact, some of your questions are actually discussed in another paper also written by us, Lee et al. (2018), titled "Determining variabilities of non-Gaussian wind-speed distributions using different metrics and timescales". This is a complementary project of this manuscript, and we examine the results from different spread and distribution metrics with data of different averaging timescales. In short, different metrics should be tested regardless of the underlying wind-speed distribution, and in this manuscript, we

conclude that RCoV is the most applicable in most locations and timescales. Please visit http://iopscience.iop.org/article/10.1088/1742-6596/1037/7/072038 for more details.

References

Archer, C. L. and Jacobson, M. Z.: Geographical and seasonal variability of the global "practical" wind resources, Appl. Geogr., 45, 119–130, doi:10.1016/j.apgeog.2013.07.006, 2013.

Lee, J. C.-Y., Fields, M. J., Lundquist, J. K. and Lunacek, M.: Determining variabilities of non-Gaussian wind-speed distributions using different metrics and timescales, J. Phys. Conf. Ser., 1037(7), 072038, doi:10.1088/1742-6596/1037/7/072038, 2018.

Lunacek, M., Jason Fields, M., Craig, A., Lee, J. C. Y., Meissner, J., Philips, C., Sheng, S. and King, R.: Understanding Biases in Pre-Construction Estimates, J. Phys. Conf. Ser., 1037(6), 062009, doi:10.1088/1742-6596/1037/6/062009, 2018.

**Fig. 1.** Histogram of derived energy data among the 349 R2-filtered sites

**Fig. 2.** Updated Figure 2 in the manuscript

---

## Author Comment (AC2) · 4 Oct 2018

Because the font and font size is fixed (in plain text) as set by WES, for clarity, the reviewer comments are numbered, and the first paragraphs of our responses open with "——".

(1) Excellent work and paper. My minor comment is with respect to the use of reanalysis data which may have a lower interannual variability than actual site data. A short comment by the authors in the paper could address this quite easily.

—— We thank the reviewer for the suggestion. Lines 209 to 211 now read: "The MERRA-2 data of coarse temporal and spatial resolutions may also represent a lower intermonthly or IAV than the wind sites actually experience."